# Specific depletion of the motor protein KIF5B leads to deficits in dendritic transport, synaptic plasticity and memory

Junjun Zhao[1†], Albert Hiu Ka Fok[1†], Ruolin Fan[1], Pui-Yi Kwan[1], Hei-Lok Chan[1], Louisa Hoi-Ying Lo[1], Ying-Shing Chan[2], Wing-Ho Yung[3], Jiandong Huang[1,2,4], Cora Sau Wan Lai[1,2]*, Kwok-On Lai[1,2]*

[1]School of Biomedical Sciences, The University of Hong Kong, Hong Kong, China; [2]State Key Laboratory of Brain and Cognitive Sciences, The University of Hong Kong, Hong Kong, China; [3]School of Biomedical Sciences, Chinese University of Hong Kong, Hong Kong, China; [4]Institute of Synthetic Biology, Shenzhen Institutes of Advanced Technology, Chinese Academy of Sciences, Shenzhen, China

**Abstract** The kinesin I family of motor proteins are crucial for axonal transport, but their roles in dendritic transport and postsynaptic function are not well-defined. Gene duplication and subsequent diversification give rise to three homologous kinesin I proteins (KIF5A, KIF5B and KIF5C) in vertebrates, but it is not clear whether and how they exhibit functional specificity. Here we show that knockdown of KIF5A or KIF5B differentially affects excitatory synapses and dendritic transport in hippocampal neurons. The functional specificities of the two kinesins are determined by their diverse carboxyl-termini, where arginine methylation occurs in KIF5B and regulates its function. KIF5B conditional knockout mice exhibit deficits in dendritic spine morphogenesis, synaptic plasticity and memory formation. Our findings provide insights into how expansion of the kinesin I family during evolution leads to diversification and specialization of motor proteins in regulating postsynaptic function.

**\*For correspondence:**
coraswl@hku.hk (CSWL);
laiko@hku.hk (K-OL)

[†]These authors contributed equally to this work

**Competing interests:** The authors declare that no competing interests exist.

## Introduction

Synapse maturation and remodeling are crucial for brain functions including learning and memory. The postsynaptic sites of excitatory synapses are located on the dendritic spines, which undergo dynamic structural changes that are essential for experience-driven wiring of the neuronal network (*Trachtenberg et al., 2002*). More than 1000 proteins with diverse structures and functions have been identified in the postsynaptic density (PSD) (*Bayés et al., 2011*), and a tight regulation of their abundance and localization is essential for proper synapse development and plasticity. Many of the postsynaptic proteins are locally translated in dendrites, which allows spatial and temporal regulation of molecular composition of individual synapses in response to local extracellular stimuli (*Holt and Schuman, 2013*). To achieve protein synthesis in dendrites, mRNAs synthesized in the soma need to be assembled in ribonucleoproteins (RNPs) and transported over long distances by molecular motors along microtubule (*Doyle and Kiebler, 2011*).

Kinesin and dynein superfamilies of proteins are microtubule-dependent molecular motors that mediate long-distance transport of materials in neuron. The kinesin superfamily is very diverse and contains 45 members in mammal. It is sub-divided into 14 different families based on structural similarity (*Hirokawa et al., 2010*). The kinesin I family (encoded by the *Kif5* genes) contains the founding kinesin protein kinesin heavy chain (KHC) (*Brady, 1985*; *Vale et al., 1985*). While only one single KIF5 is present in invertebrates such as *Drosophila*, *C. elegans* and *Aplysia*, gene duplication events give rise to three homologous KIF5 genes (*Kif5a*, *Kif5b* and *Kif5c*) in vertebrates (*Miki et al., 2001*).

**eLife digest** Transporting molecules within a cell becomes a daunting task when the cell is a neuron, with fibers called axons and dendrites that can stretch as long as a meter. Neurons use many different molecules to send messages across the body and store memories in the brain. If the right molecules cannot be delivered along the length of nerve cells, connections to neighboring neurons may decay, which may impair learning and memory.

Motor proteins are responsible for transporting molecules within cells. Kinesins are a type of motor protein that typically transports materials from the body of a neuron to the cell's periphery, including the dendrites, which is where a neuron receives messages from other nerve cells. Each cell has up to 45 different kinesin motors, but it is not known whether each one performs a distinct task or if they have overlapping roles.

Now, Zhao, Fok et al. have studied two similar kinesins, called KIF5A and KIF5B, in rodent neurons to determine their roles. First, it was shown that both proteins were found at dendritic spines, which are small outgrowths on dendrites where contact with other cells occurs. Next, KIF5A and KIF5B were depleted, one at a time, from neurons extracted from a brain region called the hippocampus. Removing KIF5B interfered with the formation of dendritic spines, but removing KIF5A did not have an effect. Dendritic spines are essential for learning and memory, so several behavioral tests were conducted on mice that had been genetically modified to express less KIF5B in the forebrain. These tests revealed that the mice performed poorly in tasks that tested their memory recall.

This work opens a new area of research studying the specific roles of different kinesin motor proteins in nerve cells. This could have important implications because certain kinesin motor proteins such as KIF5A are known to be defective in some inherited neurodegenerative diseases.

Unlike KIF5B which is ubiquitously expressed, KIF5A and KIF5C are mostly expressed in neuron (*Kanai et al., 2000*). Functional redundancy has been demonstrated among the three KIF5s, as exogenous expression of KIF5A or KIF5C can rescue the impaired mitochondrial transport in cells lacking KIF5B (*Kanai et al., 2000*). In contrast, specific function of individual KIF5 has been reported in zebrafish, in which axonal transport of mitochondria depends only on KIF5A but not the other two KIF5s (*Campbell et al., 2014*). Furthermore, only KIF5A dysfunction leads to seizure and the neuro-muscular disorder Hereditary Spastic Paraplegia (*Fink, 2013*; *Nakajima et al., 2012*). It is therefore plausible that the expansion of the *Kif5* gene family during evolution enables functional specificity of individual KIF5 in the vertebrate brain, although the molecular basis of the specificity has not been identified. The three KIF5s contain motor, stalk, and tail domains (*Friedman and Vale, 1999*), and they all bind to kinesin light chain (KLC) which mediates interaction with some of the cargoes (*Kamal et al., 2000*; *Morfini et al., 2016*). Despite the overall structural similarity, the carboxyl-termini (starting from around amino acid 934 until the last amino acid) of the three KIF5s are very different, which may confer the individual KIF5 distinctive functions in neurons.

Previous studies have mostly focused on KIF5 function in axonal transport because the motor domain of KIF5 preferentially moves out of dendrites into axons, and KIF5 function is negatively regulated by the dendritic protein MAP2 (*Gumy et al., 2017*; *Huang and Banker, 2012*; *Kapitein et al., 2010*; *Tas et al., 2017*). However, all three KIF5s are co-purified with RNPs, and dominant-negative KIF5 disrupts the dendritic localization of RNA-binding proteins (*Kanai et al., 2004*). Additional dendritic cargoes for KIF5, including the AMPA receptor/GRIP1 complex and GABA$_A$ receptor, have also been identified (*Heisler et al., 2014*; *Nakajima et al., 2012*; *Setou et al., 2002*; *Twelvetrees et al., 2010*). KIF5s therefore likely participate in both axonal and dendritic transport. Despite previous studies on its importance on AMPA receptor trafficking (*Kim and Lisman, 2001*; *Setou et al., 2002*; *Hoerndli et al., 2013*; *Heisler et al., 2014*), the role of KIF5 on dendritic spine morphogenesis and synaptic plasticity has not been comprehensively examined. In this study, we aim to investigate whether the three KIF5s have specific roles in the development and function of excitatory synapses on the postsynaptic neuron, and what might underlie the functional specificity.

Here we report that KIF5B but not KIF5A is specifically involved in the development of excitatory synapses of postsynaptic neurons and dendritic transport of the RNA-binding protein fragile X mental retardation protein (FMRP). The diverse carboxyl-termini of KIF5A and KIF5B determine their functional specificity, and we further identified arginine methylation of KIF5B as a novel post-translational modification (PTM) in regulating cargo binding. Because of the embryonic lethality of KIF5B knockout mice that precludes their use to study the synaptic and cognitive functions of adult brain in vivo, we generate mice with KIF5B conditional knockout in CaMKIIα-expressing neurons. The KIF5B conditional knockout mice exhibit altered dendritic spine structural plasticity in vivo, as well as deficits in synaptic plasticity and memory formation. Our study strongly suggests that homologous motor proteins of the kinesin I family have non-redundant functions in regulating the development and function of excitatory synapses that is crucial for learning and memory.

## Results

### Expression and subcellular localization of KIF5s in hippocampus

To compare the synaptic functions of different KIF5s, we mainly focus on neurons from the hippocampus, a brain region that is important for learning and memory and where the development of excitatory synapses is well-studied. We first determined the expression of different KIF5s in the hippocampus along development. Although KIF5C was previous reported to be expressed exclusively in medulla and spinal cord (*Kanai et al., 2000*), *Kif5c* mRNA is detected in the developing hippocampus in Allen Brain Atlas. Expression data for *Kif5a* and *Kif5b* transcripts in the developing brain is not available, but transcripts encoding the three KIF5s are detected in the adult mouse hippocampus in the atlas. Previous study has reported that *Kif5* mRNAs expression is unchanged in cultured hippocampal neurons along maturation in vitro (*Silverman et al., 2010*). On the other hand, we found that all three KIF5 proteins showed similar developmental expression profiles in the hippocampus, with the expression more prominent at early postnatal stages and significantly reduced at later postnatal and adult stages (*Figure 1A*). Next, we examined the distribution of KIF5 protein in the brain by fractionation. All three KIF5s were detected in the synaptic plasma membrane fraction (*Figure 1B*), which is consistent with the proteomic study reporting the presence of three KIF5s in the PSD (*Bayés et al., 2011*).

### KIF5A and KIF5B have distinct functions in excitatory synapse development and function

Many functional studies on KIF5s employ over-expression of dominant-negative constructs, which contain cargo-binding domains of the kinesin but lacking motor domains, thereby disrupting cargo movement through competitive binding. Here we attempt to address the role of individual KIF5 by specifically depleting each KIF5 homolog in neurons using RNA-interference. Three short hairpin RNAs (shRNAs) were created that specifically targeted KIF5A, KIF5B, and KIF5C. The knockdown efficiency and specificity of each shRNA in neuron were confirmed by Western blot and immunofluorescence staining (*Figure 1—figure supplement 1*). To examine the effect on excitatory synaptic transmission, whole-cell patch recording was performed in hippocampal neurons transfected with shRNAs targeting different KIF5s together with GFP construct. We found that knockdown of individual KIF5 differentially affected excitatory synaptic transmission. Compared to control shRNA, knockdown of KIF5B resulted in the most profound and significant reduction in the frequency of miniature excitatory synaptic current (mEPSC), while knockdown of KIF5C did not affect mEPSC frequency or amplitude. Notably, introduction of KIF5A-shRNA did not change the mEPSC frequency but instead significantly increased the mEPSC amplitude (*Figure 1C*).

Since the shRNA and GFP constructs were introduced to the neurons using calcium phosphate precipitation which has very low transfection efficiency, the reduction of mEPSC frequency in the GFP-positive neuron was likely due to cell-autonomous decrease in synapse number on the postsynaptic neuron instead of change in presynaptic release. To test this hypothesis, the density of different types of dendritic spines was examined. Although knockdown of either KIF5B or KIF5C caused a significant reduction in the density of mushroom spines, only the introduction of KIF5B-shRNA increased the density of filopodia. On the other hand, knockdown of KIF5A did not cause any change in the density of mushroom spines or filopodia when compared to control neurons

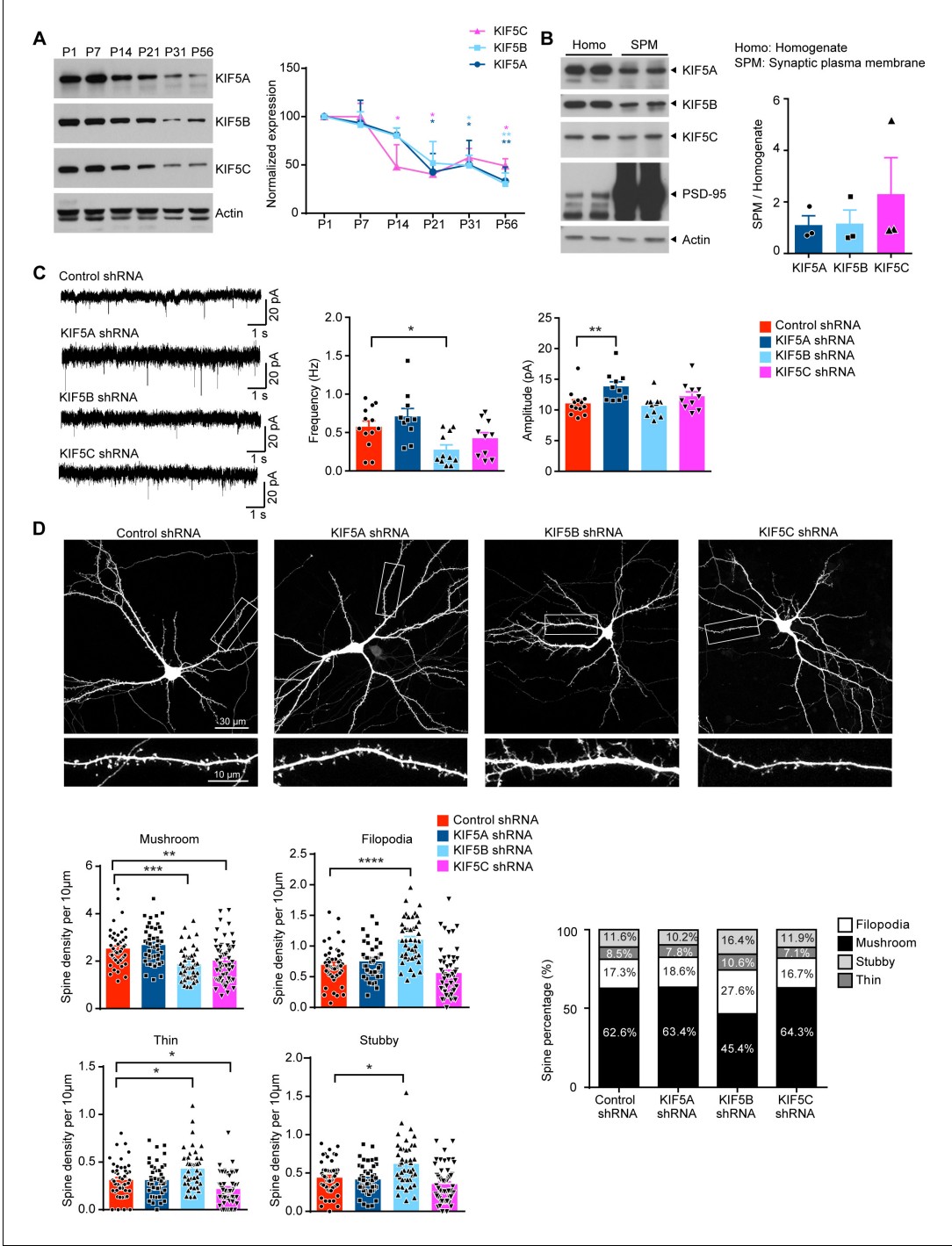

**Figure 1.** Different roles of KIF5s on dendritic spine morphogenesis and synaptic function. (**A**) Western blot showing the developmental expression of different Kinesin I motors in the mouse hippocampus. The expression of all three KIF5s significantly decreased at later postnatal stages as compared to postnatal day 1 (P1); three independent experiments, mean + SEM; *p<0.05; **p<0.01; Two way ANOVA with Dunnett's multiple comparisons test. (**B**) Presence of all three KIF5s in the synaptic plasma membrane (SPM). Postsynaptic density-95 (PSD-95) served as the positive control for the SPM fraction. (**C**) Whole-cell patch recording was performed on hippocampal neuron upon individually knocking down KIF5A, KIF5B or KIF5C with short hairpin RNAs (shRNAs). Representative traces of miniature excitatory postsynaptic currents (mEPSCs) were shown. KIF5B knockdown led to significant decrease in mEPSC frequency compared to control. KIF5A knock-down significantly increased mEPSC amplitude compared to control (n = 10–13 neurons for each condition from five independent experiments; mean

*Figure 1 continued on next page*

*Figure 1 continued*

+ SEM; *p<0.05; **p<0.01; Kruskal-Wallis test). (**D**) Representative images of dissociated rat hippocampal neurons co-transfected with GFP and the shRNA targeting individual KIF5 or a control shRNA. Neurons were transfected at days in vitro (DIV) 13, and fixed and stained by GFP antibody at DIV 16. Knockdown of KIF5B significantly reduced the density of mushroom spines and led to a significant increase in the number of filopodia, thin and stubby spines. Knockdown of KIF5C significantly decreased the number of mushroom and thin spines (42–55 neurons of each group from three independent experiments were quantified; mean + SEM; *p<0.05, **p<0.01, ***p<0.001, ****p<0.0001; Kruskal-Wallis test).

The online version of this article includes the following source data and figure supplement(s) for figure 1:

**Source data 1.** Plotted values for *Figure 1*.
**Figure supplement 1.** Validation of the knockdown efficiency and specificity of different shRNAs targeting KIF5.
**Figure supplement 1—source data 1.** Plotted values for *Figure 1—figure supplement 1*.

(*Figure 1D*). The differential effect of KIF5A and KIF5B knockdown on spine morphogenesis and synaptic transmission is not attributed to differences in knockdown efficiency, as either shRNA reduced the target KIF5 expression by similar levels (*Figure 1—figure supplement 1*). Taken together, knockdown of KIF5B in hippocampal neurons leads to more profound changes in mEPSC and dendritic spine morphogenesis than knockdown of KIF5C, while knockdown of KIF5A has no effect on dendritic spines.

To confirm that KIF5A and KIF5B indeed differentially regulate dendritic spine morphogenesis and to exclude potential off-target effect of the KIF5B-shRNA, rescue experiments using different KIF5s were performed. We focus on mushroom spines instead of the other three spine types in subsequent experiments because mushroom spines are regarded as mature spines that are more stable and possess the excitatory PSD (*Bourne and Harris, 2007*; *Berry and Nedivi, 2017*). Moreover, among the different spine types only mushroom spines were reduced after KIF5B knockdown, and the fewer mushroom spines correlated well with the decrease in mEPSC frequency. As expected, co-expression of KIF5B reversed the loss of mushroom spines induced by the KIF5B-shRNA. However, co-expression of KIF5A with the KIF5B-shRNA failed to rescue the loss of mushroom spines (*Figure 2A*). In contrast, co-expression of KIF5C fully reversed the mushroom spine defects induced by the KIF5B-shRNA (*Figure 2B*), suggesting that KIF5B and KIF5C share similar function on excitatory synapse development. Both endogenous and exogenously expressed KIF5A and KIF5B were present in dendrites and dendritic spines, and the percentage of dendritic spines containing endogenous KIF5A was even higher than that of KIF5B (*Figure 2—figure supplement 1*). These findings indicate that KIF5A and KIF5B have intrinsically distinct functions on excitatory synapses, although both KIF5A and KIF5B can be found in dendritic spines.

## Differential functions of KIF5A and KIF5B in dendritic transport of FMRP

KIF5 protein structure is divided into three domains: a motor domain, two coiled-coil domains which together form the stalk, and the tail domain (*Friedman and Vale, 1999*). Since the carboxyl termini, the most diverse regions between the KIF5s, represent part of the cargo-binding tail domain (*Morfini et al., 2016*; *Nakajima et al., 2012*) (*Figure 3A*), we next ask whether the three KIF5s might bind to cargoes differentially. We examine several different dendritic cargoes including the RNA-binding proteins (RBPs) FMRP and Ras GTPase-activating protein-binding protein (G3BP1 and G3BP2), which have been shown to regulate dendritic spine maturation (*Dictenberg et al., 2008*), as well as the AMPA receptor subunit GluA2. Pull-down assay using carboxyl-terminal fragments of individual KIF5s revealed that FMRP was preferentially pulled down by KIF5B and KIF5C but not KIF5A, while all three KIF5s could pull down G3BPs and GluA2 (*Figure 3B*).

Next, we examined whether knockdown of KIF5A and KIF5B differentially affects the dendritic localization and transport of FMRP. Neurons were co-transfected with GFP-FMRP and tdTomato, which labels the dendritic arbors and spines, together with the control shRNA, KIF5A-shRNA, or KIF5B-shRNA, followed by spinning disk confocal live imaging. Consistent with previous study on the trafficking of RBPs (*Mitsumori et al., 2017*), most FMRP granules were either stationary or exhibiting oscillatory movement, while a small proportion showing unidirectional or bidirectional movement. Compared to control shRNA, knockdown of KIF5B significantly reduced the density of FMRP

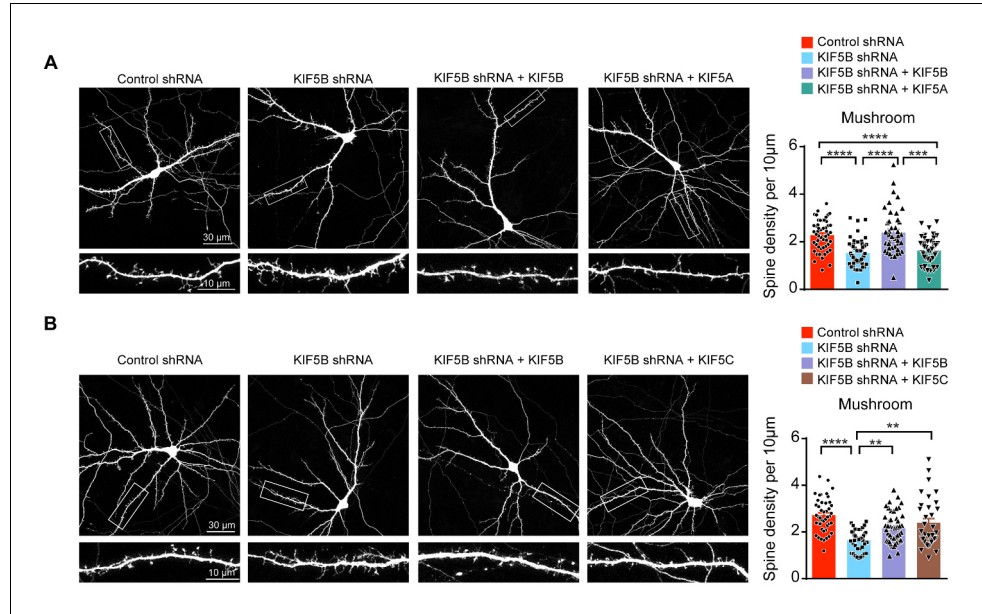

**Figure 2.** Non-redundant roles of KIF5A and KIF5B on dendritic spine morphogenesis. (**A**) Representative images of dissociated rat hippocampal neurons co-transfected with GFP and KIF5B-shRNA with/without KIF5B or KIF5A construct. Co-expression of KIF5B rescued the loss of mushroom spines induced by KIF5B-shRNA, while KIF5A expression failed to reverse the spine phenotypes (42–51 neurons of each group from three independent experiments were quantified; mean + SEM; ***p<0.001, ****p<0.0001; Kruskal-Wallis test). (**B**) Representative images of dissociated rat hippocampal neurons co-transfected with GFP and KIF5B-shRNA with/without KIF5B or KIF5C construct. Co-expression of KIF5C rescued the loss of mushroom spines induced by KIF5B-shRNA (34–46 neurons of each group from four independent experiments were quantified; mean + SEM; **p<0.01, ****p<0.0001; Kruskal-Wallis test).

The online version of this article includes the following source data and figure supplement(s) for figure 2:

**Source data 1.** Plotted values for *Figure 2*.
**Figure supplement 1.** Subcellular localization of KIF5A and KIF5B in hippocampal neurons.
**Figure supplement 1—source data 1.** Plotted values for *Figure 2—figure supplement 1*.

granules on dendrites. Interestingly, KIF5B shRNA only significantly decreased the density of stationary but not motile granules. In contrast, knockdown of KIF5A caused a general increase in the density of motile granules while decreasing the stationary granules, resulting in no net change in the density of total granules (*Figure 3C–D*). There was no effect on the motility of the unidirectional and bidirectional granules after knocking down either KIF5A or KIF5B (*Figure 3—figure supplement 1*). To further characterize the effect on FMRP function in dendrite, the localization of two FMRP-cargoes, CaMKIIα and Grin2b mRNAs, was examined using fluorescent in situ hybridization (FISH) upon knockdown of KIF5A or KIF5B, and the distribution of mRNA puncta along individual dendrites was analyzed. Consistent with the reduced density of GFP-FMRP granules, knockdown of KIF5B also significantly reduced the density of both CaMKIIα and Grin2b mRNA puncta on dendrites (*Figure 3E*). In contrast, knockdown of KIF5A did not affect CaMKIIα and Grin2b mRNA density on dendrite. Together these findings indicate that KIF5A and KIF5B differentially regulate the dendritic transport of FMRP and its mRNA cargoes.

## Carboxyl termini of KIF5A and KIF5B determine their functional specificity in neuron

What is the molecular basis of the functional specialization of KIF5A and KIF5B? The presence of a longer carboxyl-terminus in KIF5A which is very diverse from the corresponding regions of KIF5B and KIF5C (*Figure 3A*) prompt us to explore if it represents an inhibitory constraint for cargo binding. Towards this end, we created a truncated KIF5A construct with the carboxyl-terminal lacking the

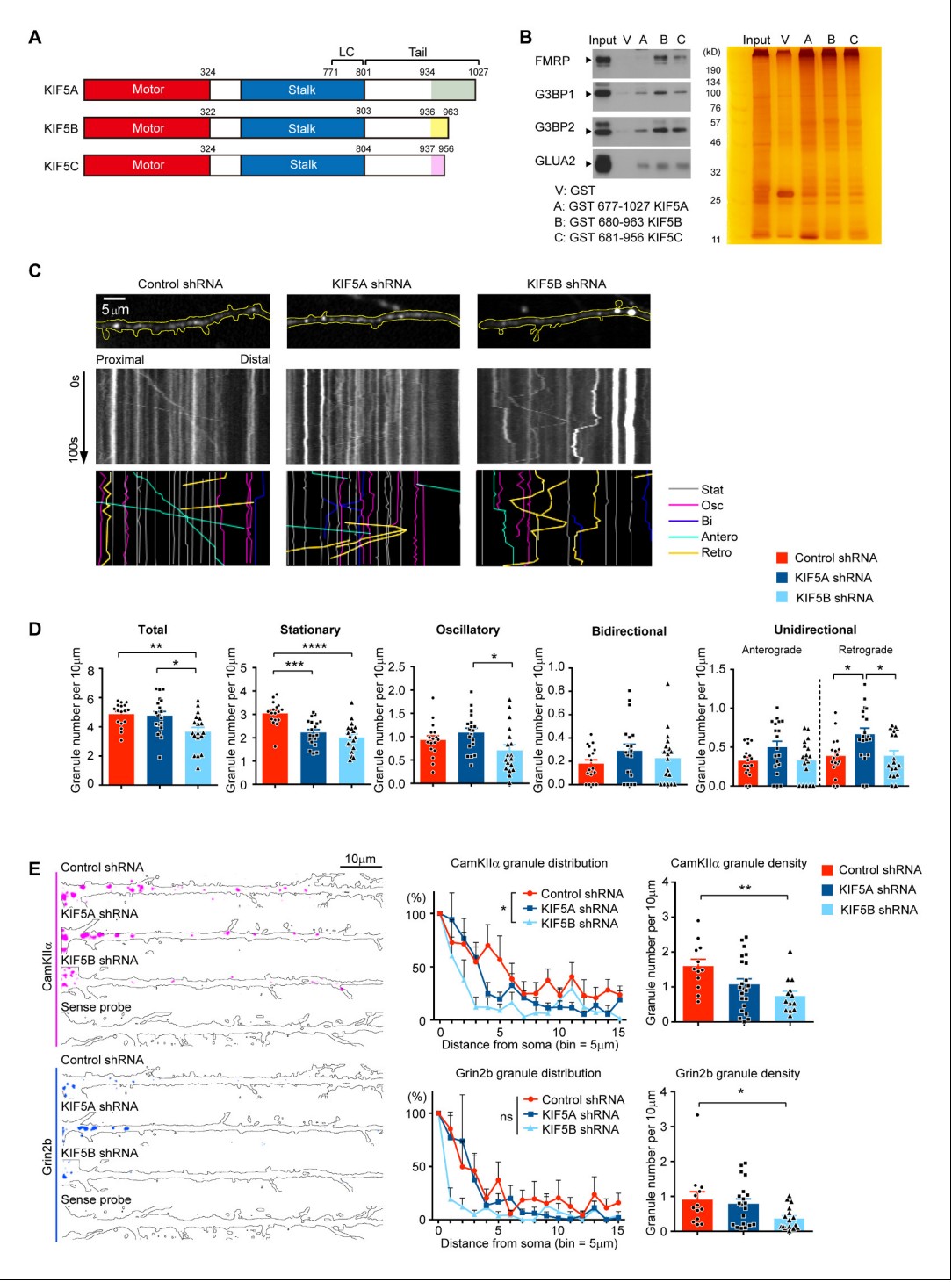

**Figure 3.** KIF5A and KIF5B differentially regulate the dendritic transport of FMRP and mRNA cargoes. (**A**) Schematic diagram of the motor, stalk and tail domains of KIF5A, KIF5B and KIF5C, with the corresponding amino-acid positions indicated. The diverse carboxyl-termini within the tail domains were colored. (**B**) Pull-down of proteins from brain homogenate by GST-tagged KIF5s. FMRP was preferentially pulled-down from the brain lysate by KIF5B and KIF5C, but not KIF5A. (**C**) Representative dendrites and kymographs of GFP-FMRP granules from each group of neurons transfected with control shRNA, KIF5A-shRNA or KIF5B-shRNA. Different types of movements were traced manually and displayed in different colors (Stat: stationary; Osc: oscillatory; Bi: bidirectional; Antero: anterograde; Retro: retrograde.). Live imaging was conducted at one frame per second for 100 s. (**D**) Quantification of the densities of granules showing each category of movement. 17–19 neurons were

*Figure 3 continued on next page*

*Figure 3 continued*

quantified for each experimental condition. Results were pooled from three independent experiments; mean + SEM; *p<0.05, **p<0.01, ***p<0.001, ****p<0.0001, one-way ANOVA, Tukey's multiple comparisons test for total, stationary, oscillatory and anterograde granules; Kruskal-Wallis, Dunn's multiple comparisons test for bidirectional and retrograde granules. (E) Left: representative FISH images of CaMKIIα and Grin2b mRNA puncta along dendrites. Hybridization with the sense probe served as negative control for the in situ hybridization. Middle: distribution of puncta number along dendrites (one bin = 5 µm) from cell body. 12–18 neurons were quantified for Grin2b mRNA and 11–20 neurons were quantified for CaMKIIα mRNA; mean + SEM; *p<0.05, two-way ANOVA, Tukey's multiple comparisons test. Right: quantification of granule density. 12–22 neurons were quantified for CaMKIIα mRNA analysis; mean + SEM; **p<0.01, one-way ANOVA, Tukey's multiple comparisons test. 13–20 neurons were quantified for Grin2b mRNA; mean + SEM; *p<0.05, one-way ANOVA, Kruskal-Wallis test. Results were pooled from three independent experiments.

The online version of this article includes the following source data and figure supplement(s) for figure 3:

**Source data 1.** Plotted values for *Figure 3*.
**Figure supplement 1.** Quantification of GFP-FMRP granule motility and percentage of movements after knockdown of KIF5A or KIF5B.
**Figure supplement 1—source data 1.** Plotted values for *Figure 3—figure supplement 1*.

---

last 88 amino acids, as well as a chimeric KIF5A in which the last 88 amino acids were substituted by the shorter carboxyl-terminus of KIF5B. Either one of these constructs but not the wild-type KIF5A was able to pull down FMRP from the synaptoneurosome (SNS), suggesting that the carboxyl-terminus of KIF5A indeed inhibits binding of specific cargoes (*Figure 4A*). Remarkably, when shRNA targeting KIF5B was introduced into hippocampal neurons to induce loss of mushroom spines, co-expression of the chimeric KIF5A that contained the carboxyl-terminus of KIF5B was able to reverse the spine phenotype (*Figure 4B*). These findings indicate that the last 88 amino acids of KIF5A

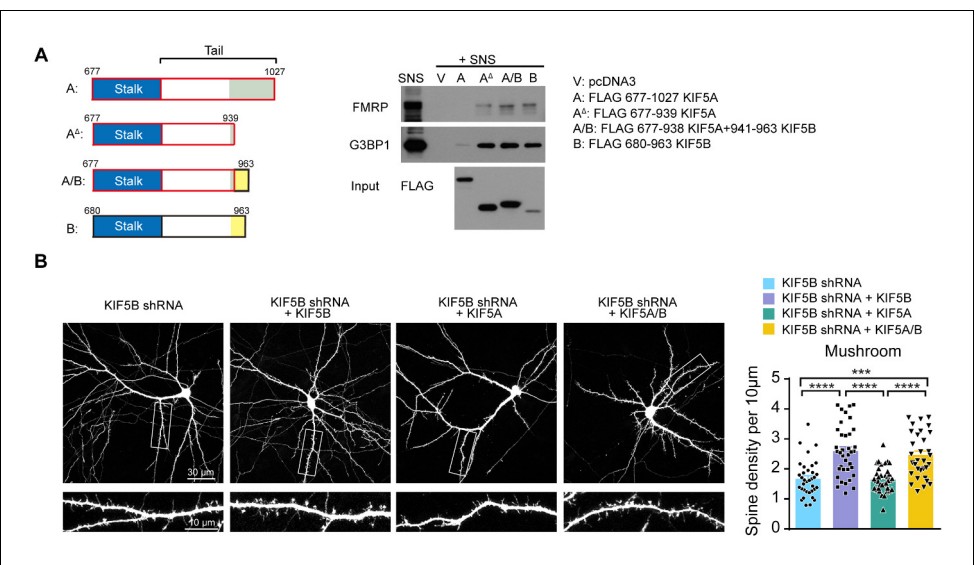

**Figure 4.** Carboxyl-terminus determines the functional specificity of KIF5A and KIF5B in dendritic spine morphogenesis. (**A**) Schematic diagram of the different constructs for pull-down assay (left panel). FMRP and G3BP1 were pulled-down from the SNS by either the truncated KIF5A lacking the 88 amino acids at the carboxyl-terminus, or chimeric KIF5A/B in which the carboxyl-terminus was substituted by that of KIF5B (right panel). (**B**) Representative images of dissociated rat hippocampal neurons co-transfected with GFP and KIF5B-shRNA with/without KIF5B, KIF5A or chimeric KIF5A/B containing the carboxyl-terminus of KIF5B (1–938 KIF5A+941–963 KIF5B). Co-expression of chimeric KIF5A/B rescued the loss of mushroom spines induced by KIF5B-shRNA (32–37 neurons of each group from three independent experiments were quantified; mean + SEM; ***p<0.001, ****p<0.0001; Kruskal-Wallis test).

The online version of this article includes the following source data for figure 4:

**Source data 1.** Plotted values for *Figure 4*.

prevent the motor protein to promote dendritic spine maturation, while its substitution by the shorter carboxyl terminus of KIF5B is sufficient to regain its synaptic function.

## Arginine methylation near the carboxyl- terminus of KIF5B is required for its synaptic function

Amino acid sequence alignment of the carboxyl termini of different KIF5s revealed the presence of two arginine residues (Arg-941 and Arg-956) followed by glycine residues (the RGG motif) in KIF5B that are conserved across different vertebrates. KIF5C contains only the Arg-941 but not Arg-956, while these two RGG motifs are absent in the KIF5A carboxyl-terminus (*Figure 5A*). The RGG motifs often undergo arginine methylation, which involves the addition of methyl group to the guanidine nitrogen atom of arginine and is catalyzed by the protein arginine methyltransferases (PRMT) (*Najbauer et al., 1993*). Hundreds of arginine-methylated proteins in the adult mouse brain have recently been identified by mass spectrometry (*Guo et al., 2014*), and our data mining results indicated that KIF5B was one of the methylated proteins. Although arginine methylation is a well-established mechanism in the regulation of gene transcription and splicing in the nucleus (*Bedford and Clarke, 2009*), emerging studies have indicated their function outside the nucleus, in particular their importance in synaptic functions (*Penney et al., 2017*). We therefore investigate whether arginine methylation represents a novel post-translational mechanism in regulating kinesin functions. We first confirmed the arginine methylation of KIF5B and KIF5C but not KIF5A when exogenously expressed in 293 T cells (*Figure 5B*). Using reciprocal immunoprecipitation with antibodies that recognize the mono-arginine methylation within glycine-rich region or KIF5B, we confirmed that KIF5B was methylated in the synaptoneurosome (*Figure 5C*). To determine whether the two conserved RGG sequences within the carboxyl-terminus of KIF5B are indeed the major methylation sites, we substituted the two arginine residues to histidine by site-directed mutagenesis, which retained the positive charges of the residues but could not undergo PRMT-mediated methylation. The KIF5B R941H or R956H mutant showed reduced methylation, whereas arginine methylation was absent in the double mutant (R941/956H) in which both arginine residues were substituted by histidine (*Figure 5D*). These results indicate that R-941 and R-956 are the two major methylation sites of KIF5B.

To ask whether and how arginine methylation affects KIF5B function, pull-down experiments were performed using the wild-type or methylation-deficient mutant (R941/956H) of KIF5B. The amount of FMRP and G3BP1 pulled down by the methylation-deficient mutant was significantly reduced when compared to wild-type KIF5B (*Figure 5E*). To address whether arginine methylation is required for the synaptic function of KIF5B, we first compared the activity of wild-type and methylation-deficient mutant in the formation of mushroom spines using the KIF5B-shRNA rescue experiments. Co-expression of wild-type KIF5B reversed the loss of mushroom spines induced by the knockdown of KIF5B, while the methylation-deficient KIF5B failed to rescue the mushroom spine loss (*Figure 5F*). Moreover, co-expression of wild-type but not the methylation-deficient KIF5B with the KIF5B-shRNA significantly increased the mEPSC frequency, (*Figure 5G*). These results are consistent with the hypothesis that arginine methylation at the carboxyl-terminus is essential for KIF5B function on dendritic spine development and synaptic transmission, and suggesting a mechanism through regulating cargo-binding.

## Generation of KIF5B conditional knockout mice

Since KIF5B homozygous knockout is embryonic lethal (*Tanaka et al., 1998*), we generated a KIF5B conditional knockout (*Kif5b$^{-/-}$*) mice using the Cre/loxP gene-targeting strategy to study the function of KIF5B in vivo. CaMKIIα promoter-driven Cre transgenic line (*CaMKIIα-Cre*) (*Tsien et al., 1996*) and *Kif5b$^{fl/fl}$* mice (*Cui et al., 2011*) were used to generate heterozygous (*CaMKIIα-Cre;Kif5b$^{fl/+}$*, Hetero) and homozygous (*CaMKIIα-Cre;Kif5b$^{fl/fl}$*, Homo) conditional knockout mice in CaMKIIα-expressing neurons, which started the expression of Cre-recombinase after birth (*Dragatsis and Zeitlin, 2000*; *Tsien et al., 1996*) (*Figure 6A*). Both homozygous and heterozygous mice were viable, and the homozygous mice did not differ in the general appearance or brain size from the wild-type (*Figure 6—figure supplement 1*). Analysis of whole-brain lysate showed a significant reduction of KIF5B protein level in homozygous knockout mice when compared to wild-type, and importantly there were no significant changes in the expression of KIF5A and KIF5C (*Figure 6B*), or the dendritic kinesin KIF17 which is crucial for synaptic plasticity and memory formation (*Song et al., 2009*;

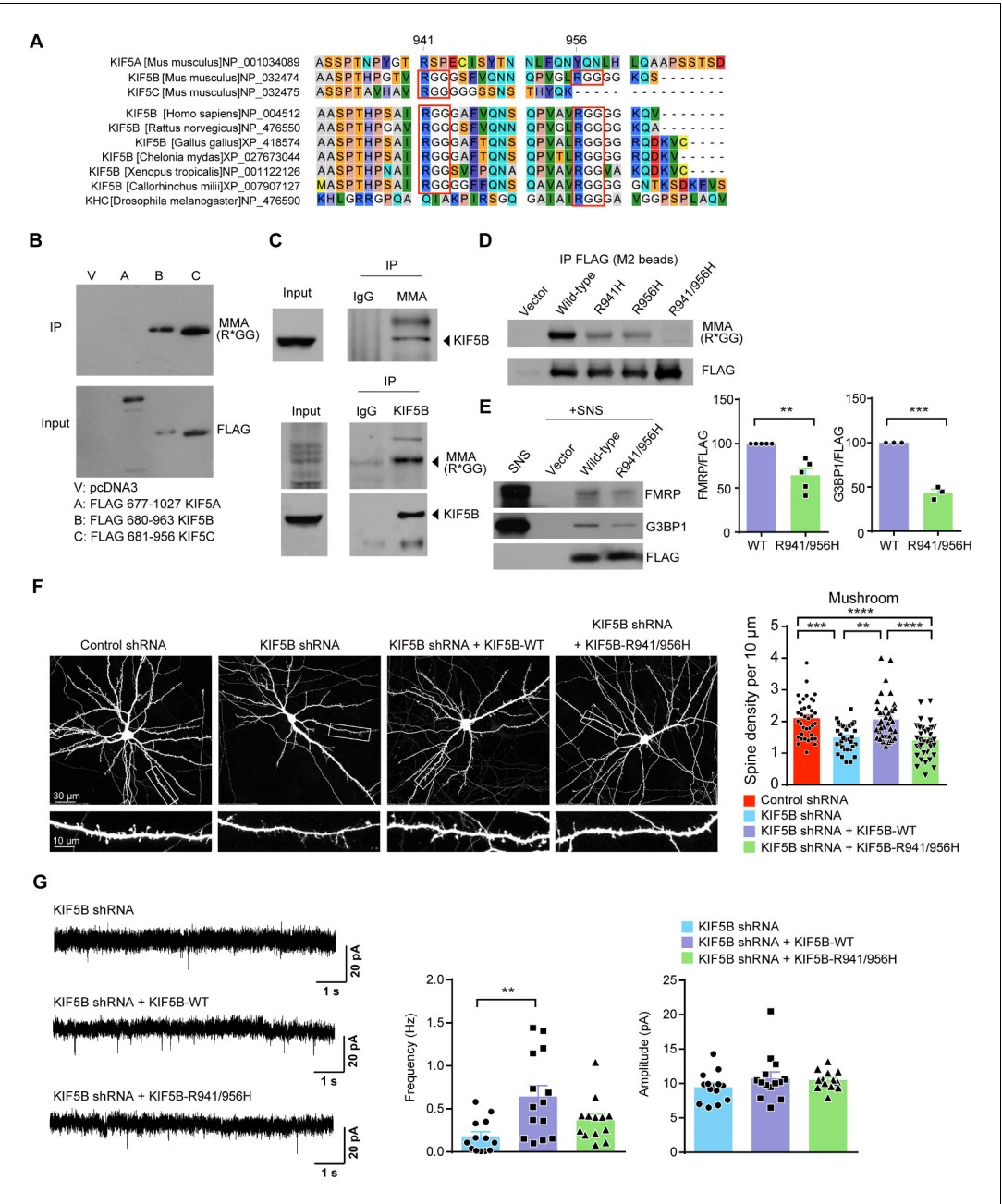

**Figure 5.** Mono-methylation of arginine near carboxyl-terminus of KIF5B is required for the formation of mushroom spines. (**A**) Amino acid alignment of the carboxyl-termini of KIF5A, KIF5B and KIF5C. Two conserved arginine residues (R941 and R956) in KIF5B across different vertebrates were highlighted by red boxes. (**B**) Carboxyl-terminal portions of different KIF5s were expressed in 293 T cells. After immunoprecipitation with anti-FLAG beads, proteins were immunoblotted by antibody targeting mono-arginine methylation at glycine-rich motifs [MMA (R*GG)]. Transfection with pcDNA3 (vector) served as a negative control. Only KIF5B and KIF5C but not KIF5A were arginine-methylated. (**C**) Methylation of KIF5B in the synaptoneurosome. Proteins were immunoprecipitated with anti-Mono-Methyl-Arginine antibody (MMA) and immunoblotted with KIF5B antibody (upper panel). Reciprocal immunoprecipitation using KIF5B antibody was performed to verify the arginine methylation pattern (lower panel). (**D**) Methylation-deficient mutants of KIF5B were constructed by single or double substitution of the two arginine residues to histidine. Vector and FLAG-tagged KIF5B constructs were transfected into 293 T cells. After immunoprecipitation with anti-FLAG beads, proteins were immunoblotted by antibody targeting mono-arginine methylation at glycine-rich motifs [MMA (R*GG)]. Mono-methylation of KIF5B was abolished by substitution of both arginine sites by histidine (R941H/R956H). (**E**) Reduced amount of FMRP and
*Figure 5 continued on next page*

*Figure 5 continued*

G3BP1 was pulled down by methylation-deficient (R941/956H) mutant of KIF5B (five independent experiments for FMRP and three independent experiments for G3BP1; mean + SEM; **p<0.01; ***p<0.001; Student's *t*-test). (**F**) Representative images of dissociated rat hippocampal neurons co-transfected with GFP and KIF5B-shRNA together with either RNAi-resistant wild-type or the methylation-deficient (R941/956H) KIF5B construct. Co-expression of wild-type but not the methylation-deficient mutant KIF5B rescued the loss of mushroom spines induced by the KIF5B-shRNA (35–38 neurons of each group from three independent experiments were quantified; mean + SEM; **p<0.01, ***p<0.001, ****p<0.0001; Kruskal-Wallis test). (**G**) Representative traces of mEPSCs of KIF5B-shRNA co-expressed with RNAi-resistant wild-type or the methylation-deficient (R941/956H) KIF5B construct were shown. Co-expression of wild-type KIF5B, but not methylation-deficient KIF5B, reversed the reduction of mEPSC frequency caused by KIF5B knock-down (13–14 neurons from five independent experiments were quantified for each experimental condition; mean + SEM; **p<0.01; Kruskal-Wallis test).

The online version of this article includes the following source data for figure 5:

**Source data 1.** Plotted values for *Figure 5*.

*Yin et al., 2011*; *Franker et al., 2016*) (*Figure 6—figure supplement 2*). We also examined the levels of KIF5B expression by immunohistochemistry in excitatory neurons using neurogranin (NRGN) as a marker in the neocortex. We found that homozygous mice showed a significant reduction of cells that were positive for both KIF5B and NRGN in the frontal association cortex (FrA) when compared to wild-type mice, without significant change in the number of neurons in this region (*Figure 6C*, *Figure 6—figure supplement 1*).

## KIF5B regulates dendritic spine density and plasticity in vivo

To determine the effect of KIF5B knockout on dendritic spines in adult neurons, the conditional knockout mice were crossbred with *Thy1*-YFP H line mice to enable sparse neuronal labeling for isolated dendrite imaging, followed by three-dimensional reconstruction for the analysis of spine number (*Figure 7A*). Conditional knockout of KIF5B at postnatal stages resulted in a significant reduction of dendritic spines in CA1 hippocampal neurons of homozygous mice (*Figure 7B*). However, the effect of KIF5B on spine density is region-specific, since the dendritic spine number was not significantly different between control and knockout mice in neurons of the FrA (*Figure 7C*). To examine the excitatory synaptic transmission of CA1 hippocampal neurons, whole-cell patch recording was conducted on hippocampal slices from the wild-type and KIF5B conditional knockout mice. CA1 hippocampal neurons of the KIF5B conditional knockout mice showed a significant reduction in both the frequency and amplitude of mEPSC as compared with wild-type neurons (*Figure 7D*). Therefore, the KIF5B conditional knockout mice showed a reduction of dendritic spine density that is associated with deficient excitatory synaptic transmission in hippocampal neurons.

Although there was no significant difference in terms of dendritic spine density in FrA in homozygous conditional knockout, this region was chosen to examine dendritic spine plasticity based on its involvement in associative learning and accessibility for in vivo transcranial imaging (*Lai et al., 2012*; *Nakayama et al., 2015*). Using two-photon microscopy, we monitored the baseline dendritic spine plasticity of adolescent mice (P31 ±1) over 7 days. Imaging sessions were performed on Day 0, 2, and 7 (*Figure 7E–F*). We found that both heterozygous and homozygous mice showed a significant increase in dendritic spine elimination compared to wild-type mice over 2 days (*Figure 7G*). However, when we examined the spine plasticity in the next time window from Day 2 to Day 7 over 5 days, both heterozygous and homozygous mice showed an increase in dendritic spine formation (*Figure 7H*). Overall, both heterozygous and homozygous KIF5B conditional knockout mice showed an increase of dendritic spine turnover rate when compared to wild-type, but only that in homozygous was statistically significant (*Figure 7I*). Although we did not observe significant difference in the survival rate of newly formed spines (*Figure 7—figure supplement 1*), we found that the increase in spine formation during the second time window was caused by the significant increase in re-formation of spines in close proximity to eliminated spines from first time window (*Figure 7J*). These data suggest that KIF5B knockout in excitatory pyramidal neurons alters normal dendritic spine plasticity with an increase of synaptic instability in the neural circuitry.

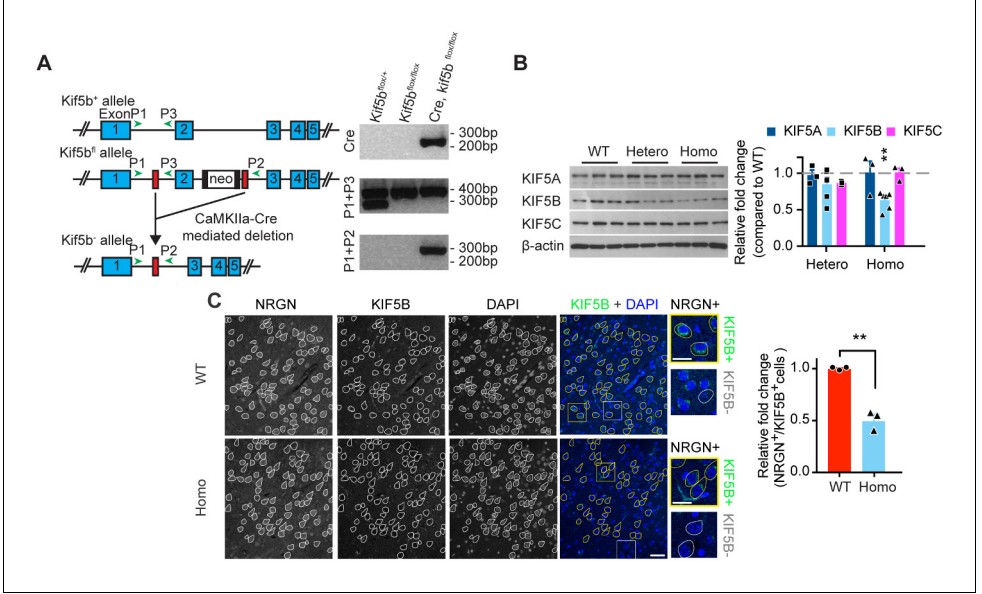

**Figure 6.** Targeting of *Kif5b* gene in *CaMKII-Cre/Kif5b^fl/fl* mice and validation of the conditional knockout mice. (**A**) Scheme of KIF5B knockout strategy (left panel). The blue rectangles (E1, E2, E3, E4, and E5) annotated exon 1, exon 2, exon 3, exon 4, and exon 5 of the *Kif5b* gene, respectively. The white rectangle with black bars on each side (each bar representing a flippase recognition target site) represented the 1.7 kb frt-NeoR-frt cassette. The red rectangles represent the LoxP sites (drawing not to scale). The green arrowheads indicated the designated annealing positions of genotyping PCR primers. Genotyping analysis of PCR product from mice DNA with different genotypes of the *Kif5b* gene (right panel). Cre primers were used to detect Cre recombinase gene. Primers P1 and P3 were used to identify *Kif5b^fl/+* and *Kif5b^fl/fl* genotypes. Primers P1 and P2 were used to identify *Kif5b* conditional knockout, *Kif5b^-/-* genotype. (**B**) Western blot analysis probed with anti-KIF5A, anti-KIF5B, anti-KIF5C and anti-β-actin antibodies. Quantification result in the right panel was presented as the relative fold change compared to wild-type (WT). n = 6 for all groups for the analysis of KIF5B. n = 3 for the analysis of other protein targets. Data were presented in mean + SEM. **p<0.01, One-way ANOVA with post hoc Tukey's HSD. (**C**) Immunohistochemical staining of sagittal brain sections showed a significant reduction of cells that were both positive for KIF5B and Neurogranin (NRGN) in frontal area on postnatal day (P) 42 ± 1. White traces highlighted NRGN-positive cells. Yellow traces indicated cells that were positive for both NRGN and KIF5B. Yellow and white squares indicate the zoom-in areas of NRGN⁺/KIF5B⁺ cells and NRGN⁺/KIF5B⁻ cells, respectively. Scale bar, 50 µm and 25 µm in magnified inserts. Right panel: quantification of KIF5B knockout in immunohistochemical staining of frontal area. n = 3 for WT. n = 3 for Homo. Data were presented in mean + SEM. **p<0.01,Student's *t*-test with Welch's correction.

The online version of this article includes the following source data and figure supplement(s) for figure 6:

**Source data 1.** Plotted values for *Figure 6*.

**Figure supplement 1.** KIF5B conditional knockout mice show normal general appearance and cortical layer architecture.

**Figure supplement 1—source data 1.** Plotted values for *Figure 6—figure supplement 1*.

**Figure supplement 2.** KIF5B conditional knockout mice show no change in KIF17 expression.

**Figure supplement 2—source data 1.** Plotted values for *Figure 6—figure supplement 2*.

## KIF5B conditional knockout mice exhibit deficits in synaptic plasticity, learning and memory

Based on the role of KIF5B on dendritic spine density and plasticity, we next investigated the impact of KIF5B conditional knockout on animal behavior. A series of behavioral tests were performed, including open field test, elevated plus maze, marble burying test, 3-chamber social interaction test, novel object recognition test, auditory-cued fear conditioning, and Barnes maze. We found that there was no significant difference in open field test, elevated plus maze, and marble burying test in heterozygous and homozygous mice when compared to wild-type, indicating that conditional knock-out of KIF5B did not lead to hyperactivity, anxiety-like or repetitive behaviors (*Figure 8—figure supplement 1*). On the other hand, homozygous mice exhibited memory deficits in a variety of learning-

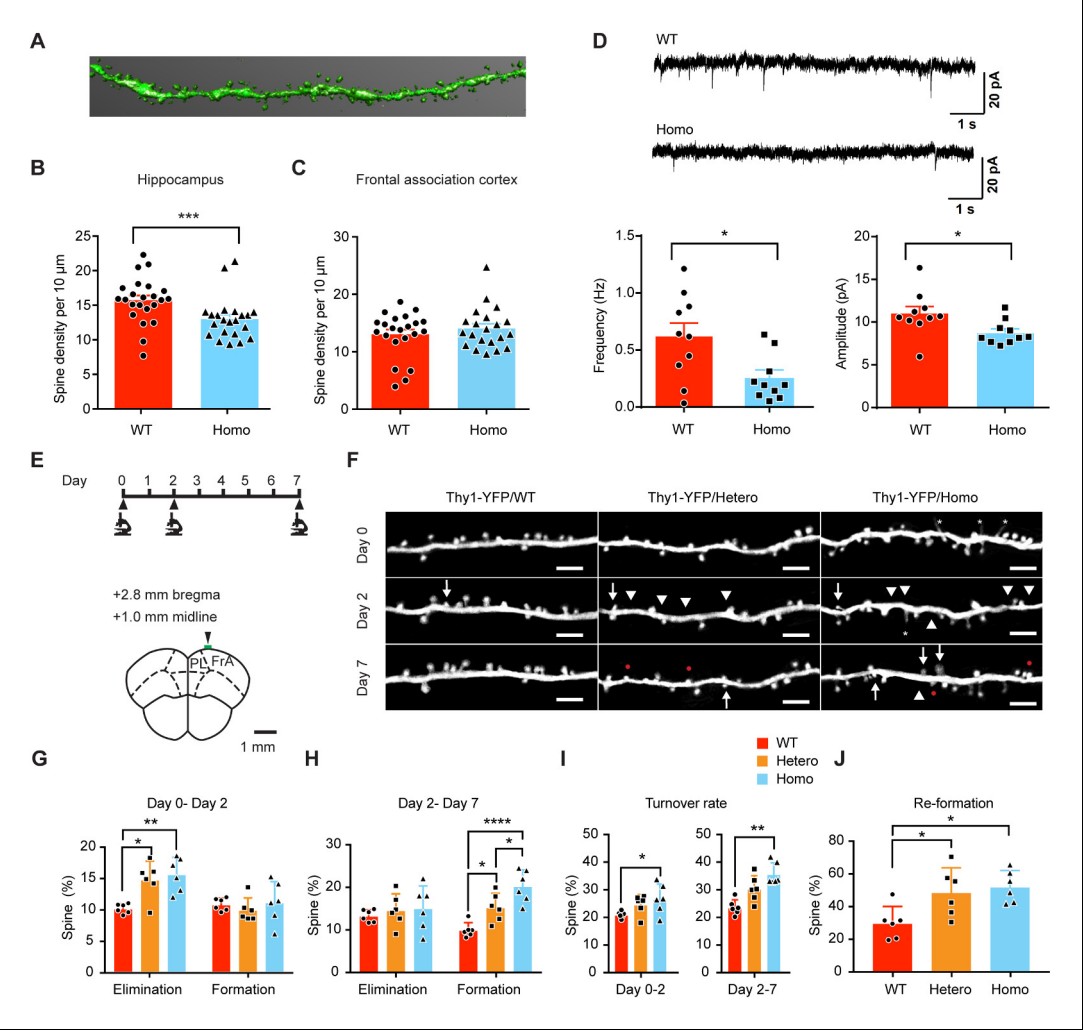

**Figure 7.** KIF5B conditional knockout mice show increase of dendritic spine instability in the frontal association cortex. (**A**) Hippocampal and frontal association cortex slices of *Thy1*-YFP;*CaMKIIα*-Cre conditional *Kif5b*<sup>fl/fl</sup> knockout mice (Homo) and *Thy1*-YFP;*Kif5b* wild-type mice (WT) at postnatal day (P) 44 were fixed. Confocal images of secondary dendrites from apical branches of CA1 hippocampal neurons and prefrontal cortex neurons were captured. 3D reconstruction of individual dendrites was performed for quantification. Representative image of a hippocampal dendrite after 3D reconstruction was shown. (**B**) The density of dendritic spines in the homozygous KIF5B conditional knockout mice was significantly reduced in CA1 hippocampus as compared to the WT mice (23 dendrites from 2 WT mice and 22 dendrites from two homo mice; mean + SEM; ***p<0.001; Mann-Whitney test). (**C**) No change in spine density was observed for neurons in the frontal association cortex (22 dendrites from 2 WT mice and 21 dendrites from two homo mice; mean + SEM; not significant; Mann-Whitney test). (**D**) Representative traces for mEPSCs on CA1 hippocampal neurons from WT and the homozygous KIF5B conditional knockout mice. The frequency and amplitude of mEPSCs from KIF5B conditional knockout neurons showed a significant reduction compared to WT neurons (10 neurons from three mice for each group; mean + SEM; *p<0.05; Student's *t*-test for mEPSC amplitude, Mann-Whitney test for mEPSC frequency). (**E**) Experimental timeline and the diagram of a coronal section of frontal association cortex (FrA) showing the imaging site (green bar). (**F**) Representative images of dendrites of *Thy1*-YFP/WT, *Thy1*-YFP/Hetero and *Thy1*-YFP/Homo at the imaging time point of Day 0, Day 2, and Day 7. Scale bars, 5 μm. Arrows mark spine formation compared to the previous time point. Arrowheads mark spine elimination compared to the previous time point. Red dots mark re-formation of previously eliminated dendritic spines in close proximity. Asterisks mark filopodia. (**G–J**) Quantification of spine elimination and formation rates from (**G**) Day 0 – Day 2, (**H**) Day 2 – Day 7, (**I**) total turnover rate and (**J**) re-formation of eliminated dendritic spines in close proximity on Day 7. n = 6, 947 dendritic spines for WT; n = 6, 906 dendritic spines for Hetero; n = 6, 1078 dendritic spines for Homo. Data were presented in mean + SD. *p<0.05. **p<0.01. ***p<0.001, One-way ANOVA for G-J, except formation in G, Day 2 – Day 7 turnover rate in I used Kruskal-Wallis test.

*Figure 7 continued on next page*

*Figure 7 continued*

The online version of this article includes the following source data and figure supplement(s) for figure 7:

**Source data 1.** Plotted values for *Figure 7*.

**Figure supplement 1.** KIF5B conditional knockout mice show no significant difference in survival rate of newly formed dendritic spines.

**Figure supplement 1—source data 1.** Plotted values for *Figure 7—figure supplement 1*.

related behaviors. In 3-chamber social interaction test, homozygous mice showed a significant reduction of social memory index (*Figure 8A*), but no significant difference in total interaction time from wild-type (*Figure 8—figure supplement 2A–B*). These data showed that KIF5B homozygous conditional knockout leads to deficits in social memory. In novel object recognition test, mice were presented with a novel object 14–16 hr after the mouse was exposed to the familiar objects for testing short-term object recognition memory. Homozygous mice showed a significantly reduced preference to the novel object (*Figure 8B*), suggesting a deficit in short-term memory recall. Next, we used auditory-cued fear conditioning to test fear associative memory. The freezing response of KIF5B conditional knockout mice was similar to wild-type in the acquisition phase (*Figure 8—figure supplement 2C*), but homozygous mice showed a significant decrease of freezing response to the conditioned stimulus (CS, auditory cue) during the recall test 48 hr after fear acquisition (*Figure 8C*). Since there was no significant difference in the trend of fear acquisition, this data indicates that homozygous mice show deficit in fear memory recall. The absence of significant deficits in heterozygous mice in these memory tests suggests the dose-dependent role of KIF5B in memory formation and retrieval. We next investigated the effect of KIF5B conditional knockout in spatial memory without using heterozygous conditional knockout mice. In Barnes maze test, mice were trained to locate the escape chamber among the 20 holes in the maze during the acquisition phase based on contextual cues. Wild-type mice showed a learning progress during the acquisition phase as indicated by a decreasing trend of primary errors they made during training, but such learning progress was not observed in homozygous mice. Homozygous mice also tended to stay in the wrong target hole instead of exploring the environment which in turn showing fewer primary errors when compared to wild-type (*Figure 8—figure supplement 2D*). Nonetheless, homozygous mice showed a significantly higher primary latency to locate the escape hole when compared to wild-type during the recall test five days after the training (*Figure 8D*).

Since the results of the behavioral tests strongly suggest deficits in hippocampal-dependent functions, we examined the expression of long-term potentiation (LTP) at the Schaffer collateral (SC)-CA1 synapses from acutely prepared hippocampal slices of the control and KIF5B homozygous conditional knockout mice. While LTP could be induced in both control and homozygous hippocampal slices, the LTP decayed faster at the homozygous SC–CA1 synapses, and the field EPSP amplitude during the last 10 min recording was significantly reduced at the homozygous SC–CA1 synapses when compared to wild-type (*Figure 8E*). There was no significant difference in input/output relationship between wild-type and homozygous mice, indicating the baseline synaptic response was not affected (*Figure 8F*). To test whether presynaptic function was altered, the pair-pulse ratios were measured with several different inter-pulse durations. We found no significant difference in pair-pulse ratio between wild-type and homozygous mice, indicating similar presynaptic responses (*Figure 8G*). Taken together, these findings suggest that conditional knockout of *Kif5b* causes memory recall deficits in social memory, object recognition memory, fear associative memory, and spatial memory, showing the important role of KIF5B in memory formation and retrieval. These memory deficits are associated with impaired long-term synaptic plasticity in the hippocampus. Moreover, the synaptic and memory deficits in the KIF5B conditional knockout mice cannot be compensated by the presence of the other two homologous KIF5s.

## Discussion

Many studies have examined the functional significance of individual kinesin through exogenous expression of dominant-negative construct, which usually contains the tail domain of the kinesin of interest without the motor domain and hence does not move along microtubule. This approach is

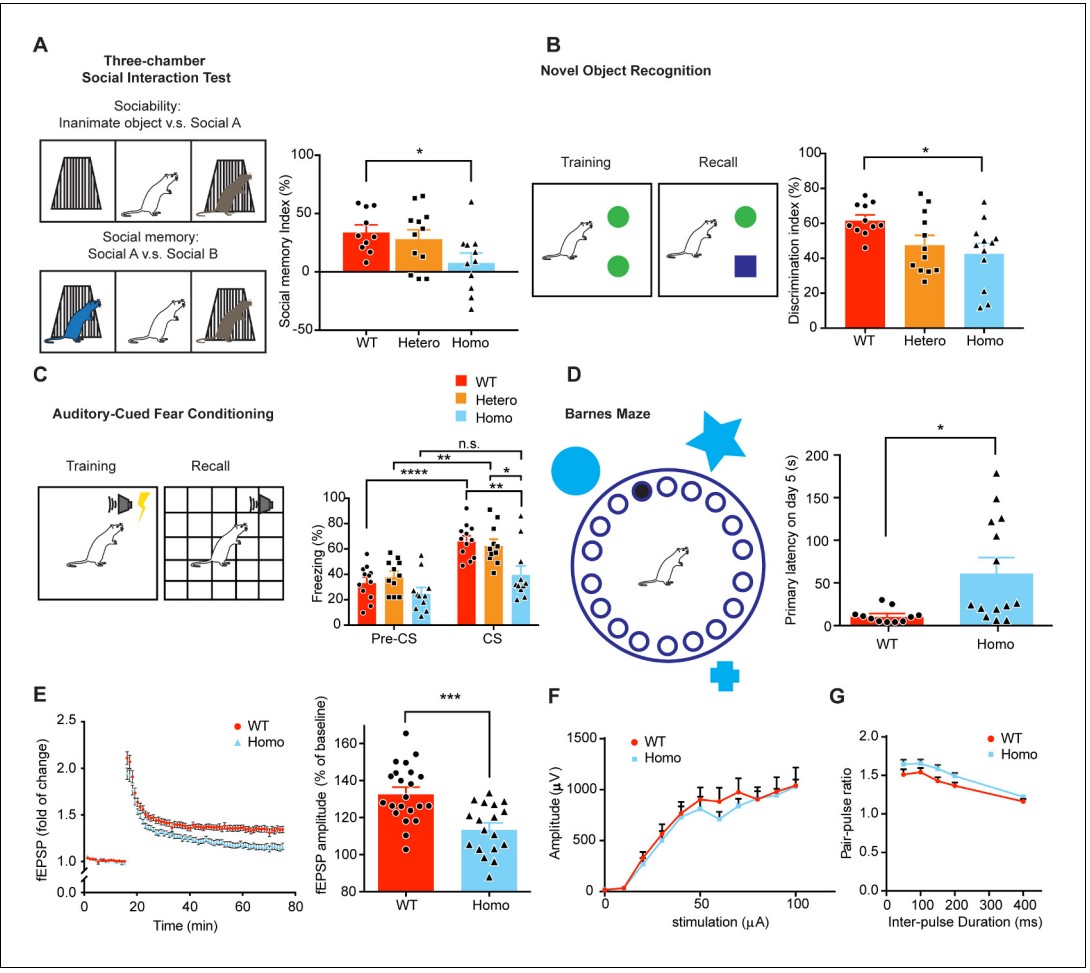

**Figure 8.** KIF5B conditional knockout mice show deficits in synaptic plasticity, learning and memory. (**A**) Schematic diagram shows the set-up of three-chamber social interaction test and the quantification of social memory index in the right panel. n = 10 for WT. n = 11 for Hetero and Homo. Data were presented in mean + SEM. *p<0.05. (**B**) Schematic diagram shows the set-up of novel object recognition test and the quantification of discrimination index during recall phase in the right panel. n = 11 for WT. n = 12 for Hetero. n = 12 for Homo. Data were presented in mean + SEM. *p<0.05. (**C**) Schematic diagram shows the set-up of fear conditioning and the quantification of freezing time before and during the recall tone was played in the right panel. n = 12 for WT. n = 11 for Hetero. n = 11 for Homo. Data were presented in mean + SEM. *p<0.05, **p<0.01, ****p<0.0001, n.s.: not significant, Two Way ANOVA with post hoc Tukey's multiple comparison test. (**D**) Schematic diagram shows the set-up of Barnes maze. Quantification of primary latency during recall on 5 days after training. n = 11 for WT. n = 13 for Homo. Data were presented in mean + SEM. *p<0.05, Mann-Whitney test. (**E**) Hippocampal LTP was reduced in KIF5B homozygous conditional knockout mice as shown by the reduced field EPSP amplitude (five mice, 18 slices) compared with wild-type (six mice, 22 slices).***p<0.001, mean + SEM, Student's *t*-test. (**F**) Input/output and (**G**) pair-pulse ratio curves from hippocampal slices of WT (three mice, 11 slices) and KIF5B homozygous conditional knockout mice (four mice, 21 slices). No significant difference between WT and Homo, two-way ANOVA.

The online version of this article includes the following source data and figure supplement(s) for figure 8:

**Source data 1.** Plotted values for *Figure 8*.

**Figure supplement 1.** KIF5B conditional knockout mice showed no significant abnormalities in anxiety-related behavior tests.

**Figure supplement 1—source data 1.** Plotted values for *Figure 8—figure supplement 1*.

**Figure supplement 2.** Sociability, fear acquisition, and primary error of Barnes maze in KIF5B conditional knockout mice.

**Figure supplement 2—source data 1.** Plotted values for *Figure 8—figure supplement 2*.

useful to demonstrate the effect of competitive binding between the dominant negative protein and endogenous motors for the cargoes. Using an alternative approach to delineate the function of individual KIF by RNAi or gene knockout, we demonstrate the important roles of KIF5B in regulating dendritic spine development and maintenance both in dissociated neurons in vitro and in the animal. Through the generation of conditional knockout mice in which the *Kif5b* gene is ablated only after birth in order to avoid lethality, we are able to demonstrate the physiological significance of KIF5B in regulating excitatory synaptic plasticity as well as learning and memory. Our findings provide compelling evidences that the function of KIF5B in neuron cannot be compensated by the other two neuron-specific KIF5s.

Since only one KIF5 is expressed in invertebrates, it appears that the neuronal-specific KIF5A and KIF5C evolve specifically for higher brain function in vertebrates. We found that knock down of either KIF5B or KIF5C, but not KIF5A, reduced mushroom spines. On the other hand, co-expression of KIF5C but not KIF5A can rescue the loss of mushroom spines caused by KIF5B-shRNA. These finding indicates that KIF5B and KIF5C share functional similarity in dendritic spine morphogenesis and their roles cannot be replaced by the functionally distinct KIF5A. However, in the KIF5B conditional knockout mice in which KIF5C expression remains unaffected, reduction in both spine density and mEPSC is observed in hippocampal neurons. Therefore, the impaired synaptic function due to KIF5B deficiency cannot be compensated by KIF5C in the postnatal brain. One possible explanation is that KIF5B is the more prominently expressed kinesin compared to KIF5C in the adult hippocampus as shown by quantitative immunoblot (*Kanai et al., 2000*). The presence of KIF5C in the conditional knockout mice may not be sufficient to compensate for the shortage of motor proteins after ~50% reduction of KIF5B expression.

The carboxyl termini of the three KIF5s share little amino acid sequence similarity. The carboxyl terminus may not bind to cargo directly since GST-KIF5B constructs without this region (a.a. 936–963) still pull down various cargoes (*Setou et al., 2002*; *Cho et al., 2007*; *Xu et al., 2010*; *Barry et al., 2014*; *Lin et al., 2019*). Our present findings also suggest that the carboxyl-terminus is not directly involved in FMRP binding because removing it (amino acid residues 939–1027) from KIF5A increases, rather than decreases, the pull-down of FMRP. Furthermore, although replacement of the KIF5A carboxyl-terminus by the KIF5B counterpart increases the binding to FMRP and G3BP1, given that the input of KIF5B is much less than the chimeric KIF5A (*Figure 4A*), it is likely that equal amount of KIF5B would pull down much more FMRP and G3BP1. This again points to the involvement of KIF5 sequence besides the carboxyl-terminus in cargo-binding. Nonetheless, in the rescue experiments with chimeric KIF5A, swapping the carboxyl-terminus with KIF5B is sufficient to transform KIF5A into a kinesin motor that enhances spine maturation, therefore unraveling a new function of the carboxyl-terminus in determining functional specficity of KIF5s. In this regard, it is noteworthy that the longer carboxyl-terminus of KIF5A binds directly to GABA$_A$ receptor-associated protein for the development of inhibitory synapses (*Nakajima et al., 2012*). Our findings together raise the interesting possibility that there is a division of labor among the two KIF5s in regulating excitatory and inhibitory synapses, and the evolution of their diverse carboxyl-termini confer them functional specificities.

Many axonal cargoes, such as syntabulin, SNAP25 and amyloid precursor protein (APP) have been identified for KIF5s (*Hirokawa and Tanaka, 2015*; *Kamal et al., 2000*). KIF5 motor domain also predominately recognizes axonal rather than dendritic microtubules, which highlight its functional significance in axon (*Kapitein et al., 2010*). However, KIF5 is also implicated in the transport of cargoes such as GABA$_A$ receptor (*Twelvetrees et al., 2010*; *Nakajima et al., 2012*), AMPA receptor (*Heisler et al., 2014*; *Setou et al., 2002*) and RNPs (*Kanai et al., 2004*), which are believed to be mainly carried to the dendrites of mature neurons. We found that KIF5B is localized not only in the axons, but is also present in the dendrites and dendritic spines of dissociated hippocampal neurons, supporting the role of dendritic KIF5B in the development of excitatory postsynaptic sites. Although it was originally thought that microtubule is not present in dendritic spines, emerging study has revealed the invasion of microtubule and kinesin to the spine heads from dendritic shaft, which are crucial for dendritic spine plasticity (*Jaworski et al., 2009*; *McVicker et al., 2016*). Our findings suggest that KIF5B might represent one of the kinesin motors that deliver synaptic proteins to the dendritic spines.

Dendritic spines exist as heterogeneous morphologies, which are usually classified into short stubby spines with no apparent spine neck, thin spines with elongated necks and small heads,

mushroom-shaped spines with large bulbous heads, and filopodia which are long and thin and do not possess a PSD (*Ziv and Smith, 1996*; *Bourne and Harris, 2007*; *Lai and Ip, 2013*; *Berry and Nedivi, 2017*). Stubby and filopodia are regarded as immature dendritic protrusions because they are relatively scarce in the mature brain (*Harris et al., 1992*). The distinct morphologies are critical to determine the properties and functions of dendritic spines. These include signal compartmentalization, calcium dynamics, capacity of local translation, and turnover (*McKinney, 2010*). Mushroom spines possess larger PSD which are correlated with greater synaptic strength and stability for information storage; while the dynamic thin spines are transient, but they may become persistent in response to a learning paradigm and contribute to the remodeling of neural circuits (*Bourne and Harris, 2007*; *Berry and Nedivi, 2017*). It is interesting that knockdown of KIF5B specifically decreases mushroom spines in cultured hippocampal neurons while increasing the abundance of the other three types of spines. Emerging studies have demonstrated that different spine types can be regulated differentially and independently (*Sanders et al., 2012*; *Spiga et al., 2014*). At the molecular level, we have also identified the postsynaptic scaffolding protein STRN4, which is encoded by a dendritic mRNA and its expression depends on NMDA receptor activity, is involved specifically in the maintenance of mushroom spines (*Lin et al., 2017*). It is tempting to speculate that a subset of proteins and/or mRNAs may depend on KIF5B for the delivery to mushroom spines that confer their selective maintenance.

Since KIF5s can pull down RNPs from the brain (*Kanai et al., 2004*), one possible mechanism by which KIF5B promotes the maintenance of mushroom spines is through the dendritic transport of mRNAs and RNA-binding proteins. We have found that knockdown of KIF5B reduced the dendritic localization of FMRP and two associated RNA transcripts as compared to knockdown of KIF5A, indicating their differential functions in dendritic transport of mRNAs. This may explain the altered spine morphology after knockdown of KIF5B, since the depletion of FMRP in mouse brain also resulted in an increase of dendritic filopodia (*Comery et al., 1997*). The local translation of CaMKIIα and Grin2b mRNAs is critical to synaptic plasticity (*Kuklin et al., 2017*; *Williams et al., 2007*), which may contribute to the disrupted LTP in mouse hippocampus upon KIF5B depletion. FMRP and associated mRNA transport involves interaction with KLC (*Dictenberg et al., 2008*). Since both KIF5A and KIF5B contain the conserved KLC binding domain, there could be additional mechanism that underlies the specific role of KIF5B in FMRP transport, which may involve the preferential interaction between FMRP and the KIF5B tail domain as revealed by our pull-down assay. It is also intriguing that KIF5B-shRNA only leads to fewer stationary granules on dendrites without affecting the motile oscillatory, unidirectional and bidirectional granules. Since other kinesins besides KIF5 can also bind to FMRP (*Charalambous et al., 2013*; *Davidovic et al., 2007*), we speculate that different pools of FMRP granules are carried by different KIFs, with KIF5B mainly responsible for the less motile granules while other KIFs transport the more motile pools of FMRP. It was recently reported that different KIFs transport cargoes with different velocities and MAP2 inhibits KIF5B activity in dendrites by interacting with the coiled-coil region and blocking microtubule binding (*Gumy et al., 2017*). This study therefore also suggests that KIF5B-mediated transport in dendrites is ineffective as compared to other kinesins. Alternatively, since microtubule and dynein are required for mRNA anchoring in *Drosophila* embryos (*Delanoue and Davis, 2005*), it is possible that besides a conventional transport function, KIF5s may help anchoring the dendritically localized FMRP and mRNAs near synapses for local translation in response to extracellular stimuli such as BDNF or synaptic activity (*Schratt, 2004*).

The function of kinesin is regulated by post-translational modification. Previous studies on the Kinesin-2 motor protein KIF17 revealed a novel mechanism of cargo release through calmodulin-dependent protein kinase (CaMKII)-mediated phosphorylation, which disrupts the interaction with the adaptor protein LIN10 and unloads the NMDA receptor subunit 2B (GluN2B) containing vesicles (*Guillaud et al., 2008*). On the other hand, the association between synaptotagmin-containing vesicles and the motor adaptor UNC76 of KIF5 in *Drosophila* is strengthened by phosphorylation (*Toda et al., 2008*). In the present study, we have characterized the methylation of two RGG motifs within the carboxyl-terminus of KIF5B involving Arg-941 and Arg-956. Invertebrates such as *C. elegans* have shorter carboxyl-terminus of KIF5 that lacks the RGG motif, while *Drosophila* has one RGG motif containing Arg-956, same as the mammalian KIF5C. The two RGG motifs in KIF5B are conserved across many vertebrates, indicating the importance of arginine methylation. Here we show that the KIF5B methylation is essential for the formation of mushroom spines and it modulates

the interaction of KIF5B with FMRP, therefore unraveling a previously unidentified PTM in regulating kinesin function. There are extensive cross-talks between arginine methylation and other PTMs, such as phosphorylation, ubiquitination, and acetylation (*Basso and Pennuto, 2015*; *Yang et al., 2018*). Future studies are needed to investigate how arginine methylation of KIF5B may interact with other forms of PTM in regulating cargo-binding of the motor protein.

Does KIF5B play any specific role in learning and memory? To answer this question, we generated the KIF5B conditional knockout mouse line in CaMKIIα-expressing neurons. Since the expression of CaMKIIα is developmentally regulated and is restricted to the forebrain with high levels in the pyramidal neurons of the neocortex and hippocampus (*Dragatsis and Zeitlin, 2000*; *Tsien et al., 1996*), we can specifically knockout KIF5B postnatally without affecting early neurodevelopment. Here we demonstrated that specific knockout of *Kif5b* in CaMKIIα-expressing neurons leads to deficits in memory recall in social memory, novel object recognition, auditory-cued fear conditioning, and spatial memory tests, with no significant deficit during initial memory acquisition phase. Furthermore, the KIF5B conditional knockout mice show deficits in the maintenance of LTP in CA1 hippocampal neurons and the loss of dendritic spines. Although there is no significant decrease of dendritic spine density in the frontal association cortex of conditional knockout mice, the rates of dendritic spine formation and elimination are significantly higher at different time points in two-photon in vivo imaging, suggesting the increase of dendritic spine instability in this region. Increase in dendritic spine instability has been commonly found in various disease models, such as Fragile X syndrome (*Nagaoka et al., 2016*; *Pan et al., 2010*), schizophrenia (*Fénelon et al., 2013*), spinocerebellar ataxia type 1 (*Hatanaka et al., 2015*) and Huntington disease (*Murmu et al., 2013*). It has been found that a small fraction of the population of transient spines grows after experience or behavioral training over days can be stabilized over the animal's lifetime, contributing to long-lasting circuit remodeling associated with new experience (*Yang et al., 2009*). The enhanced dendritic spine instability in KIF5B conditional knockout mice could contribute to brain dysfunction and deficits in learning and memory. Since the frontal cortex maturation happens at later developmental stage (*Caballero et al., 2016*; *Gogtay et al., 2004*; *Zuo et al., 2005*), the lack of dendritic spine density difference in the frontal association cortex between wild-type and KIF5B conditional knockout mice could be due to the delay of frontal cortex maturation and pruning in the conditional knockout mutant. Nonetheless, the impairments in memory recall, LTP maintenance, and dendritic spine deficits in KIF5B conditional knockout demonstrate the crucial role of KIF5B in learning and memory that cannot be compensated by KIF5A and KIF5C in vivo. The process of memory storage is not a random event. The synaptic tagging and capture hypothesis proposes that the synapses activated during LTP induction become 'tagged' (*Rogerson et al., 2014*). These tagged synapses become a target for subsequent plasticity-related product (PRP) trafficking. The capture of these PRPs by specific synapses is essential for their structural modification, as well as the maintenance of LTP and long-term memory formation. The deficits that we observed in KIF5B conditional knockout mice could be stemmed from the impairment of PRP trafficking specifically delivered by KIF5B in dendrites in response to activity-dependent plasticity.

Taken together, our findings have revealed the significance of KIF5B in regulating excitatory synapse development and function of neuron both in vitro and in vivo, and support the notion that the three homologous KIF5s have non-redundant functions in the brain. It is plausible that homologous members of the other kinesin families also exhibit functional specificity in the brain, an interesting research area which warrants further study in the future.

## Materials and methods

**Key resources table**

| Reagent type (species) or resource | Designation | Source or reference | Identifiers | Additional information |
|---|---|---|---|---|
| Strain, strain background (*M. musculus*) | C57/6J | The University of Hong Kong Laboratory Animal Unit | | |

*Continued on next page*

*Continued*

| Reagent type (species) or resource | Designation | Source or reference | Identifiers | Additional information |
|---|---|---|---|---|
| Genetic reagent (*M. musculus*) | Thy1-YFP-H | Jackson Laboratory. | 003782 \|thy1-YFP-H | |
| Genetic reagent (*M. musculus*) | CaMKIIα-Cre | Jackson Laboratory. | 005359 \|T29-1 | |
| Genetic reagent (*M. musculus*) | Kif5bfl/fl | Jiandong Huang | PMID: 20870970 | |
| Genetic reagent (*M. musculus*) | CaMKIIα-Cre;Kif5bfl/fl | This paper | | generated from breeding of CaMKIIα-Cre and Kif5bfl/fl mice |
| Transfected construct (*M. musculus*) | FLAG-677–1027 KIF5A | This paper | | aa 677–1027 of mouse KIF5A with a N-terminal FLAG tag was inserted into pcDNA3 |
| Transfected construct (*M. musculus*) | FLAG-677–939 KIF5A | This paper | | aa 677–939 of mouse KIF5A with a N-terminal FLAG tag was inserted into pcDNA3 |
| Transfected construct (*M. musculus*) | FLAG-677–938 KIF5A+941–963 KIF5B | This paper | | aa 677–938 of mouse KIF5A and aa 941–963 of mouse KIF5B with a N-terminal FLAG tag was inserted into pcDNA3 |
| Transfected construct (*M. musculus*) | FLAG-680–963 KIF5B | This paper | | aa 680–963 of mouse KIF5B with a N-terminal FLAG tag was inserted into pcDNA3 |
| Transfected construct (*R. norvegicus*) | KIF5A shRNA | This paper | | 5'-TGGAAACGCCACAGATATC-3' |
| Transfected construct (*M. musculus*) | KIF5B shRNA | This paper | | 5'-GGACAGATGAAGTATAAAT-3' |
| Transfected construct (*R. norvegicus*) | KIF5C shRNA | This paper | | 5'-GACCCTGGCAGATGTGAAT-3' |
| Transfected construct (*R. norvegicus*) | control shRNA | *Lin et al., 2017* | PMID: 28442576 | 5'-GGCTACCTCCATTTAGTGT-3' |
| Transfected construct (*M. musculus*) | pKin1A | Anthony Brown | RRID:Addgene_31607 | |
| Transfected construct (*M. musculus*) | pcDNA3-KIF5B | Jiandong Huang | PMID: 23293293 | |
| Transfected construct (*M. musculus*) | pGFP-Kif5c | Michelle Peckham | RRID:Addgene_71853 | |
| Biological sample (*R. norvegicus*) | Primary hippocampal neuron; primary cortical neuron | *Lin et al., 2017* | PMID: 28442576 | Procedures of preparing primary neurons were described in *Lin et al. (2017)* |
| Biological sample (*M. musculus*) | Synaptoneurosome | *Scheetz et al., 2000* | PMID: 10700251 | Procedures of preparing synaptoneurosome were described by *Scheetz et al. (2000)* |
| Biological sample (*M. musculus*) | Synatpic plasma membrane | *Bermejo et al., 2014* | PMID: 25226023 | Procedures of preparing synaptic plasma mebrane were described by *Bermejo et al. (2014)* |
| Antibody | KIF5B | Jiandong Huang | PMID: 20870970 | |
| Antibody | KIF5A | Abcam | RRID:AB_2132218 | |
| Antibody | KIF5C | Abcam | RRID:AB_304999 | |
| Antibody | FMRP | Abcam | RRID:AB_2278530 | |
| Antibody | KIF17 | Sigma | RRID:AB_477148 | |
| Antibody | FLAG | Sigma | RRID:AB_262044 | |
| Antibody | G3BP1 | Bethyl | RRID:AB_1576539 | |

*Continued on next page*

Continued

| Reagent type (species) or resource | Designation | Source or reference | Identifiers | Additional information |
|---|---|---|---|---|
| Antibody | G3BP2 | Bethyl | RRID:AB_1576545 | |
| Antibody | GluA2 | Millipore | RRID:AB_2113875 | |
| Antibody | RNMT | Millipore | RRID:AB_11215450 | |
| Antibody | NRGN | Millipore | Cat#AB5620 | |
| Antibody | NeuN | Millipore | Cat#AB377 | |
| Antibody | PSD-95 | NeuroMab | RRID:AB_2292909 | |
| Antibody | methylated mono-arginine R*GG | Cell Signaling | RRID:AB_10896849 | |
| Antibody | Mouse IgG2a anti-GFP | Invitrogen | RRID:AB_221568 | |
| Antibody | Rabbit anti-RFP | Rockland | RRID:AB_2209751 | |
| Antibody | Alexa 488 anti mouse IgG2a | Invitrogen | RRID:AB_2535771 | |
| Antibody | Alexa 546 anti rabbit IgG | Invitrogen | RRID:AB_2534077 | |
| Antibody | horseradish peroxidase-conjugated goat anti-rabbit IgG | Cell Signaling | RRID:AB_2099233 | |
| Antibody | horseradish peroxidase-conjugated goat anti-mouse IgG | Cell Signaling | RRID:AB_330924 | |
| Recombinant DNA reagent | pFRT-TODest FLAGHAhFMRPiso1 | Thomas Tuschl | RRID:Addgene_48690 | |
| Recombinant DNA reagent | tdTomato | Michael Davidson | RRID:Addgene_54653 | |
| Recombinant DNA reagent | GST-fused KIF5A | This paper | | aa 677–1027 of mouse KIF5A was inserted into pGEX-6P-2 |
| Recombinant DNA reagent | GST-fused KIF5B | This paper | | aa 680–963 of mouse KIF5B was inserted into pGEX-6P-2 |
| Recombinant DNA reagent | GST-fused KIF5C | This paper | | aa 681–956 of mouse KIF5C was inserted into pGEX-6P-2 |
| Recombinant DNA reagent | pEGFP-N1-KIF5A | This paper | | Constructed by inserting PCR-amplified mouse KIF5A coding sequences into the pEGFP-N1 plasmid using KpnI and BamHI |
| Recombinant DNA reagent | pEGFP-N1-KIF5B | This paper | | Constructed by inserting PCR-amplified mouse KIF5B coding sequences into the pEGFP-N1 plasmid using KpnI and BamHI |
| Sequenced-based reagent | Grin2b transcript probe (NM_012574.1, type 1) | ThermoFisher | Cat#VC1-16464 | |
| Sequenced-based reagent | Type one sense probe | ThermoFisher | Cat#VC1-20903 | |
| Sequenced-based reagent | CaMKIIa transcript probe (NM_012920.1, type 6) | ThermoFisher | Cat#VC6-11639 | |
| Sequenced-based reagent | Type six sense probe | ThermoFisher | Cat#VC6-16372 | |
| Chemical compound, drug | FLAG beads | Sigma | RRID:AB_10063035 | |
| Chemical compound, drug | glutathione sepharose four fast flow beads | GE Healthcare | Cat#17-5132-01 | |

*Continued*

| Reagent type (species) or resource | Designation | Source or reference | Identifiers | Additional information |
|---|---|---|---|---|
| Chemical compound, drug | Protein A-Sepharose beads | GE Healthcare | Cat#17-5280-01 | |
| Commercial assay or kit | Lipofectamine LTX with Plus Reagent | ThermoFisher Scientific | Cat#15338100 | |
| Commercial assay or kit | SilverQuest Silver Staining Kit | Life technologies | Cat#LC6070 | |
| Commercial assay or kit | ViewRNA ISH Cell Assay Kit | ThermoFisher | Cat#QVC0001 | |
| Commercial assay or kit | Neon transfection system | ThermoFisher Scientific | | Model MPK5000 |
| Software, algorithm | Volocity | Quorum Technologies | RRID:SCR_002668 | |
| Software, algorithm | Zen digital imaging software | Zeiss | RRID:SCR_013672 | |
| Software, algorithm | Actimetrics FreezeFrame software | Coulbourn Instruments | RRID:SCR_014429 | Version 2.2 |
| Software, algorithm | ANY-maze software | ANY-maze | RRID:SCR_014289 | |
| Software, algorithm | MetaMorph software | Molecular Devices | SCR_002368 | |
| Software, algorithm | GraphPad Prism | GraphPad Prism (https://graphpad.com) | RRID:SCR_015807 | Version 6 |
| Software, algorithm | FIJI | FIJI (https://imagej.net/Fiji) | RRID:SCR_002285 | |
| Software, algorithm | KymoResliceWide | Eugene Katrukha (https://github.com/ekatrukha/KymoResliceWide) | | |
| Software, algorithm | Straighten | Eva Kocsis (https://imagej.nih.gov/ij/plugins/straighten.html) | PMID: 1817611 | |
| Software, algorithm | Mini Analysis Program | Synaptosoft | RRID:SCR_002184 | |
| Software, algorithm | CLC Main Workbench | Qiagen (https://www.qiagenbioinformatics.com/products/clc-main-workbench/) | RRID:SCR_000354 | |

## Antibodies, chemicals and DNA constructs

Antibody against KIF5B was previously described (*Cui et al., 2011*), while others were purchased commercially, including antibodies against KIF5A, KIF5C, FMRP (Abcam), KIF17, FLAG (Sigma), G3BP1, G3BP2 (Bethyl), GluA2, RNMT, NRGN and NeuN (Millipore), PSD-95 (NeuroMab), methylated mono-arginine R*GG (Cell Signaling), GFP (Invitrogen), and RFP (Rockland). Alexa-conjugated secondary antibodies (Invitrogen) were used for immunofluorescence and horseradish peroxidase-conjugated goat anti-rabbit IgG or anti-mouse IgG (Cell Signaling) were used for western blot analysis.

For the specific knockdown of KIF5A, KIF5B, and KIF5C, a 19-nucleotide (KIF5A: 5'-TGGAAACGCCACAGATATC-3', KIF5B: 5'-GGACAGATGAAGTATAAAT-3', KIF5C: 5'-GACCCTGGCAGATGTGAAT-3') sequence derived from the rat KIF5A mRNA, mouse KIF5B mRNA at the 3'-UTR and rat KIF5C mRNA were used to create the shRNA constructs after subcloning into the pSUPER vector (Oligoengine). The sequence of control shRNA is 5'-GGCTACCTCCATTTAGTGT-3'. Full-length mouse KIF5A and KIF5C constructs were obtained from Quan Hao (The University of Hong Kong), and the coding sequence was amplified and subcloned into pcDNA3 backbone. Full-length mouse KIF5B was amplified by PCR using the plasmid pcDNA3-FLAG-KIF5B as template, which contains the insert of full-length mouse KIF5B coding region. Methylation-deficient R941H, R956H and R941/956H constructs were created by site-directed mutagenesis and the PCR products were digested by DpnI (NEB) at 37℃ water bath for 3 hr before transformation into *E. coli* competent cells. The nucleotide sequence was verified by Sanger sequencing. For GFP-FMRP construct,

the human FMRP coding sequence was amplified from the plasmid pFRT-TODestFLAGHAhFMR-Piso1 that was from Thomas Tuschl (Addgene #48690) and cloned into the pEGFP-C1 backbone using SacI and EcoRI. KIF5A-GFP and KIF5B-GFP were constructed by inserting PCR-amplified mouse KIF5A and KIF5B coding sequences into the pEGFP-N1 plasmid using KpnI and BamHI. All PCR in this study was performed using high-fidelity Pfu DNA polymerase (Agilent Technologies, Inc).

## Animals

Mice were group housed under a 12 hr light/dark cycle, with food and water available ad libitum. C57BL/6 mice expressing CaMKIIα-Cre and yellow fluorescent protein (YFP) in layer V pyramidal neurons (*Thy1*-YFP-H) and CaMKIIα promoter-driven Cre transgenic mice were purchased from the Jackson Laboratory. *Kif5b*^fl/fl^ mice were described previously (*Cui et al., 2011*). CaMKIIα promoter-driven Cre transgenic mice were used to conditionally delete exons flanked by loxP. Mice were then further crossed with *Thy1*-YFP-H line to allow imaging of layer V pyramidal neurons. Sample size was decided based on experiments in previous studies (*Lai et al., 2012*; *Yang et al., 2014*). For animal behavioral tests and in vivo imaging experiments, results from at least two independent experiments were pooled together for analysis. Mice were group housed in The Laboratory Animal Unit, The University of Hong Kong, accredited by Association for Assessment and Accreditation of Laboratory Animal Care International. Four to five weeks old mice were used in this study unless stated otherwise. All experiments were approved and performed in accordance with University of Hong Kong Committee on the Use of Live Animals in Teaching and Research guidelines.

## Electrophysiology

Whole-cell recordings were obtained by the MultiClamp 700B amplifier (Molecular Devices). For cultured hippocampal neurons, which were recorded at DIV 16–17, the pipettes with a resistance of 3–5 MΩ were filled with the internal solution consisting of 115 mM CsCl, 10 mM HEPES, 2 mM $MgCl_2$, 4 mM NaATP, 0.4 mM NaGTP, 0.5 mM EGTA, and pH was adjusted to 7.2–7.4 by CsOH. The neurons were perfused with the external solution of the following composition: 110 mM NaCl, 5 mM KCl, 2 mM $CaCl_2$, 0.8 mM $MgCl_2$, 10 mM HEPES, 10 mM Glucose, and pH was adjusted to 7.2–7.4 by NaOH. For miniature excitatory postsynaptic currents (mEPSCs) recording, tetrodotoxin (1 μM) and bicuculline (20 μM) were added into the external solution to block action potentials and the inhibitory current from GABA receptor, respectively. The signals were filtered at 2 kHz and sampled at 20 kHz using the Digidata 1440A (Molecular Devices). The holding potential is at −70 mV, and the recording lasts for 5 to 10 min. The data were analyzed by the commercial software MiniAnalysis (Synaptosoft).

For recording mEPSCs in dorsal hippocampal CA1 brain slices, postnatal day (P) 45 ± 3 wild-type and KIF5B conditional knockout mice were perfused by ice-cold dissection buffer (92 mM NMDG, 2.5 mM KCl, 1.25 mM $NaH_2PO_4$, 30 mM $NaHCO_3$, 25 mM glucose, 20 mM HEPES, 5 mM Na-ascorbate, 3 mM Na-pyruvate, 2 mM thiourea, 10 mM $MgSO_4$ and 0.5 mM $CaCl_2$ pH = 7.1–7.3) after euthanized. The brains were taken out immediately and submerged in ice-cold dissection buffer. Coronal brain slices containing CA1 were sectioned in 250 μm by vibratome. Slices were recovered in warm artificial cerebral spinal fluid (ACSF) at 32°C for 15 min, followed by room temperature incubation. The recordings were performed in ACSF at room temperature. The ACSF consisted of the following (in mM): 119 NaCl, 2.5 KCl, 1 $MgCl_2$, 2 $CaCl_2$, 26 $NaHCO_3$, 1.23 $NaH_2PO_4$ and 10 glucose. All solutions were oxygenated by 95% $O_2$/5% $CO_2$. Internal solution consisted of the following (in mM) 131 Cs-methanesulfonate, 20 CsCl, 8 NaCl, 10 HEPES, 2 EGTA, 2 NaATP and 0.3 NaGTP, pH7.3, osmolarity 290 mOsm. The glass micropipette was filled with internal solution (resistance 4–6 MΩ) and connected to the electrode for recording. The mEPSCs were recorded with the presence of 1 μM tetrodotoxin, 10 μM bicuculline and 1 μM strychnine.

For recording LTP, hippocampal slices from the wild-type and the KIF5B conditional knockout mice (3 months old) were prepared. A planar multi-electrode recording setup (MED64 system, Alpha Med Sciences Co., Ltd, Japan) was employed to record the field excitatory postsynaptic potential (fEPSP), and to study LTP. Briefly, hippocampal slices were placed on special probes that were fabricated with 8 × 8 electrode arrays and pre-coated with polyethylenimine (PEI, Sigma). The P210A probes (Alpha Med Sciences) with an inter-electrode distance of 100 μm were routinely used. Correct placement of the electrodes at the CA3–CA1 region was done manually, monitored by a

microscope (MIC-D, Olympus Ltd., Japan). To increase the efficiency of the experiments and to minimize the variation in the results arising from differences in incubation times, a maximum of 4 slices were studied simultaneously. Each slice was superfused by oxygenated ACSF. fEPSPs were recorded from the dendritic layer of CA1 neurons by choosing an electrode in the Schaffer collateral pathway as the stimulating electrode. Based on the stimulus–response curve, we chose a stimulation intensity that evoked the fEPSP with a magnitude of 30–40% of the maximum response. After allowing a stable baseline of 30 min, an induction protocol consisting of 1 train of 100 Hz stimulus that lasted for 1 s was applied, and the field potential response for 1 hr after the tetanus was recorded. The magnitude of the LTP was quantified as % change in the average amplitude of the fEPSP taken from 50 to 60 min interval after induction. To assess basal synaptic transmission, the input-output relationship was generated by delivering 10–100-μA electrical stimuli, and the amplitude of the peak fEPSPs was measured. To characterize the paired-pulse ratio, twin stimuli that were separated by a variable time interval (50, 100, 150, 200 or 400 ms) were delivered to the CA3-CA1 pathway ten times each, and the average ratio of the amplitude of the second evoked fEPSPs over the first one was determined. All the electrophysiology experiments were performed and analyzed blinded.

## Primary cell culture and transfection

Primary hippocampal neurons and cortical neurons were prepared from embryonic day 18–19 embryos of Sprague Dawley rats according to our previous study (*Lin et al., 2017*). Hippocampal neurons were cultured on 18 mm coverslips or 35 mm MatTek dishes (with 14 mm central glass, MatTek corp) dishes coated with poly-D-lysine (1 mg/ml, Sigma P0899) at high density ($1.4 \times 10^5$ cells per coverslip for dendritic spine analysis; $2 \times 10^5$ per cover glass on MatTek dish for live cell imaging of GFP-FMRP) or low density ($0.4 \times 10^5$ cells per coverslip for FISH and immunofluorescence staining) in Neurobasal medium supplemented with 2% B27% and 0.5% L-glutamate. Hippocampal neurons were transfected with different plasmids using calcium phosphate precipitation as previously described (*Lai et al., 2008*). Cortical neurons were transfected by electroporation using the Neon transfection system (ThermoFisher Scientific), in which a total of $1 \times 10^6$ cells in suspension were electroporated in each reaction with the parameter of 1500V pulse voltage and 20 ms pulse width. After electroporation, cells were plated on 35 mm dishes and cultured for 5 days before Western blot analysis.

## Live cell imaging and image analysis

Images were taken using Perkin Elmer UltraView Vox Spinning Disk Confocal Microscope 60x oil-immersion objective (NA 1.40) at a resolution of $512 \times 512$ pixels, one frame per second for 100 s. Images were exported using Volocity software and processed using FIJI software. Kymographs of selected dendrites were generated in FIJI software using the 'KymoResliceWide' plugin. The kymographs were randomized and reviewed blindly, and images with low signal-to-noise ratio were excluded due to the difficulty in quantification. The movement of individual granules in selected kymographs was then traced manually by drawing polygonal lines as overlays on the image and the traces were reviewed by an experimenter blind to the conditions. Minimum and maximum values of the kymographs were constantly adjusted during manual tracing due to uneven intensity on different segments of the dendrite but were limited to a range that was considered appropriate for that batch of images. The traces were then exported with information of the x and y coordinates of each point on the polygonal lines. To classify the type of movement exhibited by each granule, the net displacement (ND) and lateral maximal displacement (LMD) were measured. ND is defined as the difference in x coordinates of the first point and the last point of the trace. Lateral maximal displacement (LMD) is defined as the absolute value of maximal difference in x coordinates of all the points on the trace (the difference between the most proximal point and the most distal point). Granules with ND $\geqq 2$ μm are defined as unidirectional, within which the granules with ND >0 are defined as anterograde (from soma towards distal dendrite) and granules with ND <0 are defined as retrograde (from distal dendrite to soma). For granules with ND <2 μm, granules with LMD <1 μm are defined as stationary; granules with 1 μm$\leqq$LMD <2 μm are defined as oscillatory; while granules with LMD $\geqq$ 2 μm are defined as bidirectional. The motility of each granule in unidirectional or bidirectional movements was further quantified in terms of travel distance, maximal run length and maximal velocity. Travel distance is defined as the total length of the granule trajectory within 100 s. The maximal run length

is defined as the largest x-axis distance of a period of movement with constant velocity. The maximal velocity is defined as the maximum of all velocity values of a granule, which is calculated by dividing the x-axis distance of each segment of the trajectory by the y-axis distance (the time covered by this segment).

## Fractionation of synaptic plasma membrane (SPM) and Western blot analysis

SPM fraction was prepared using sucrose gradient method as described (*Bermejo et al., 2014*). Briefly, mouse brains (~P20) were homogenized in 0.32M HEPES-buffered sucrose solution. The homogenate was either centrifuged at 13000 rpm for 10 min (min) yielding the supernatant (Homo) for western blot analysis, or subjected to fractionation. The homogenate was centrifuged at 900 x g for 10 min (min) to remove nuclear fraction and the crude synaptosomal fraction (P2) was enriched from the supernatant using two times of centrifugation at 10000 x g for 15 min. The P2 pellet was later subjected to hypo-osmotic shock and centrifugation at 25,000 x g for 20 min to yield the Synaptosomal Membrane Fraction (P3). The obtained pellet was then resuspended and loaded to a 0.8M/1.0M/1.2M HEPES-buffered sucrose gradient and centrifuged at 150,000 x g for 2 hr, separating fractions in different layers. The SPM fraction was collected at the 1.0M/1.2M interface, further centrifuged at 160,000 x g for 30 min, and resuspended in 50 mM HEPES/2 mM EDTA solution. For Western blot analysis, homogenate and SPM samples were diluted with RIPA buffer or 50 mM HEPES/2 mM EDTA and denatured in sample buffer (5x sample buffer: 300 mM Tris-HCl buffer pH 6.8 10% (w/v) DSD, 25% (v/v) beta-mercaptoethanol, 50% (v/v) SDS, 25% (v/v) glycerol, 0.05% (w/v) bromophenol blue).

## Synaptoneurosome (SNS) preparation, immunoprecipitation and Western blot analysis

The preparation of SNS was performed as previously described with modification (*Scheetz et al., 2000*). In; brief, P15 mice were decapitated, and cerebellum together with the superficial, retinorecipient layers of the superior colliculus were removed. The rest of the brain tissues were homogenized in ice-cold homogenized buffer (5M NaCl, 1M KCl, 1M $MgSO_4$, 0.5M $CaCl_2$, 1M $KH_2PO_4$, 212.7 mM glucose, pH 7.4) supplemented with protease inhibitor cocktail (Roche). All subsequent steps were carried out at 4°C. Samples were passed through a series of nylon filters of descending pore size. The final pass was through Millipore filter with a 10 µm pore size. Samples were then centrifuged for 15 min at 1000 x g at 4°C. The supernatant was discarded, and the pellet was resuspended in 100 µl homogenization buffer for immunoprecipitation.

To test whether KIF5B was methylated in SNS, equal amount of SNS fraction lysate (800 µg) was incubated with KIF5B or mono-methyl-arginine (Cell Signaling) antibody at 4°C with rocking overnight. Immunoprecipitate was obtained with RIPA buffer after incubation with Protein A-Sepharose beads (GE Healthcare) for 1 hr in cold room with rocking. Beads were washed four times with RIPA buffer containing various protease and phosphatase inhibitors (10 µg/ml soybean trypsin inhibitor, 10 µg/ml leupeptin, 10 µg/ml aprotinin, 2 µg/ml antipain, 30 nM okadaic acid, 5 mM benzamidine, 1 mM sodium orthovanadate, 1 mM PMSF, 1 mM sodium fluoride, 100 mM beta-glycerophosphate). Proteins were eluted by boiling in sample buffer for 6 min. The eluate was collected by centrifugation at 13000 rpm for 1 min at 4°C and then subjected to SDS-PAGE and Western blot analysis. The protein extract was boiled in sample buffer for 5 min, separated by SDS-PAGE, and transferred onto PVDF membranes, followed by blocking with 5% skim milk in TBS with 0.1% Tween 20 (TBST) for 1 hr at room temperature (RT). The membrane was incubated at 4°C with primary antibody diluted in TBST containing 5% BSA overnight. After washing three times with TBST, membranes were incubated for 1 hr at RT with HRP-conjugated secondary antibody diluted in 5% skim milk in TBST. The HRP signal was detected by ECL (Thermo Scientific) and quantified by densitometry using Photoshop software.

To map the methylation sites of KIF5B, HEK-293T cells cultured in 100 mm dishes with 80% confluence were transfected with various KIF5B plasmids using Lipofectamine (ThermoFisher Scientific). Twenty-four hours after transfection, the cells were washed by ice-cold D-PBS and lysed by RIPA containing various protease and phosphatase inhibitors. Lysate was incubated at 4°C for 45 min and the cell debris was cleared by centrifugation at 13000 rpm for 10 min at 4°C. Equal amount of lysate

(1 mg) was incubated with FLAG beads (Sigma) in cold room for 1 hr with rocking. The FLAG-beads were centrifuged at 3000 x g for 1 min at 4°C and washed for three times with RIPA buffer containing various protease and phosphatase inhibitors, and proteins were eluted by boiling in sample buffer for 6 min. The eluate was collected by centrifugation at 13000 rpm for 1 min at 4°C and then subjected to SDS-PAGE and Western blot analysis.

For pull-down experiments, different FLAG-tagged segments from KIF5s were transfected into HKE293T cells using Lipofectamine (ThermoFisher Scientific). Twenty-four hours after transfection, cell lysate was collected by RIPA buffer with various protease and phosphatase inhibitors as described above. Equal amount of lysate (1 mg) was incubated with FLAG beads (Sigma) for immunoprecipitation. SNS pellet was collected and lysed with Tris buffer (20 mM Tris, 150 mM NaCl, 1 mM EDTA, 1 mM EGTA, 5 mM NaF, 0.5% NP40). Equal amount of SNS fraction (1 mg) was incubated with the immunoprecipitation from FLAG beads at 4°C with rocking overnight. FLAG beads were centrifuged at 3000 x g for 1 min at 4°C and washed for three times with Tris buffer containing various protease and phosphatase inhibitors. Protein were eluted by boiling in sample buffer for 6 min and then subjected to SDS-PAGE and Western blot analysis.

To validate KIFs expression in the KIF5B conditional knockout mice, brain lysate was obtained from mice (P44). Protein levels were determined by blotting with anti-KIF5A, anti-KIF5B, anti-KIF5C, anti-KIF17 (all 1: 1000) and anti-β-actin (1:3000) antibodies.

## GST pull-down assay

The recombinant GST-fused proteins were expressed by *E. coli* BL21 (DE3) grown in LB culture medium. Isopropyl β-D-1-thiogalactopyranoside (0.1 mM) was used to induce expression of GST-fused KIF5A (a.a.677–1027) at 28°C for 5 hr, while 0.5 mM isopropyl β-D-1-thiogalactopyranoside was used to express all other GST-fused proteins at 37°C for 3 hr. Mice (~6 week old) were sacrificed and forebrains were homogenized and lysed with Tris buffer. The brain lysate was pre-cleared by glutathione sepharose four fast flow beads (GE health) and GST proteins with rocking at 4°C for 1 hr. Equal amount of pre-cleared brain lysate and beads were incubated with equimolar GST-fused proteins at 4°C for 2 hr with end-over-end mixing. Then, the beads were washed with Tris buffer for three times. Proteins were eluted by boiling in sample buffer for 6 min and then subjected to Western blot or silver staining using SilverQuest Silver Staining Kit (Life technologies).

## Fluorescence In Situ Hybridization (FISH)

FISH was performed using ViewRNA ISH Cell Assay Kit (ThermoFisher) following manufactural instructions. In brief, cells were fixed using 4% formaldehyde for 30 min and rinsed in 1 x PBS. Cells were then treated with detergent and incubated with custom designed probe sets against Grin2b transcript (NM_012574.1, type 1) and CaMKIIα transcript (NM_012920.1, type 6) for 3–4 hr, preamplifier mix for 30 min, amplifier mix for 30 min, and label probe sets for 30 min, all in 40°C. Coverslips were washed with wash buffer for three times in between. Anti-GFP antibody was subsequently used for immunostaining.

## Immunofluorescence staining, image acquisition, and quantitative analysis

To stain GFP-transfected neurons for dendritic spine analysis, neurons were incubated with GFP antibody (1:2000) in GDB buffer at 4°C overnight. After washing three times with phosphate washing buffer (20 mM phosphate buffer and 0.5M NaCl), neurons were incubated with Alexa488-conjugated anti-mouse IgG2a secondary antibody (1:2000 diluted in GDB buffer) at RT for 1 hr, followed by washing three times by the phosphate washing buffer before mounting. For other immunocytochemistry experiments, cells were fixed by 4% PFA/4% sucrose in D-PBS for 15 min at RT. After washing with D-PBS, cells were incubated with blocking buffer (0.4% Triton X-100 (vol/vol) and 1% BSA) for 45 min at RT, and incubated with primary antibodies in blocking buffer at 4°C overnight. Cells were washed three times with washing buffer (0.02% Triton X-100% and 1% BSA in PBS), incubated with anti-mouse IgG2a Alexa 488 conjugate and anti-rabbit IgG Alexa 546 conjugate at RT for 1 hr, followed by washing twice in washing buffer and once by D-PBS before mounting with Hydromount (National Diagnosis).

Carl Zeiss LSM 700 confocal laser-scanning microscopes installed with Zen digital imaging software were used to acquire z-stack fluorescent images using a 63x oil-immersion objective (NA 1.40) with the following parameters: 1 AU or smaller pinhole, 0.5x optical zoom, scan speed 6–8, interval 0.35 µm with 16-bit dynamic range. The images from the same experiment were captured using identical acquisition settings, except for GFP or tdTomato (RFP) staining which served to visualize dendritic arbors and spines. Images from 2 to 3 coverslips were acquired for each experimental condition, and results from three independent experiments were pooled together for analysis. Sample size was decided based on experiments in previous study (*Lin et al., 2017*).

For dendritic spine quantification in dissociated hippocampal neurons, images of the whole neuron were captured by confocal microscope and assigned a random number, and dendrites with length more than 50 µm were selected by another blinded experimenter for quantification. Dendritic spines were classified based on our previous study (*Lin et al., 2017*). The length (L), head width (H) and neck width (N) of each individual spine were measured manually using the MetaMorph software. Mushroom spines were defined as those having $H/N \geqq 1.5$; stubby spines were defined as those having $H/N \leqq 1$ and $L/N \leqq 1$; thin spines had the ratio of $1 \leqq H/N < 1.5$ and $1.5 \leqq L/N \leqq 3$. Filopodia were defined as those with the ratio of $H/N < 1.2$ and $L/N > 3$. For each neuron, one to three isolated dendrites were selected and quantified, and the average spine density would be calculated. The 'n' number is defined as the number of neurons analyzed.

For quantification of KIF5A-GFP and KIF5B-GFP puncta, images after maximal projection of multiple z-layers were intensity-adjusted to the same minimum and maximum values using FIJI software 'Brightness/Contrast' function before manual counting of puncta.

To quantify the localization of endogenous KIF5A and KIF5B by immunostaining, images after maximal projection of z-layers were intensity-adjusted to remove signals below the threshold, which is determined by the negative control without primary antibody as reference. The spine density and percentage of puncta-positive spines were quantified by manual counting.

For quantification of KIF5A and KIF5B knockdown efficiency by immunofluorescence, areas of cell soma were outlined based on GFP signals on images after maximal projection of multiple layers. The signal intensity of the target protein within selected area was measured using FIJI software.

For quantification of FISH images, selected dendrites from maximal projected images were straightened using 'Straighten' (*Kocsis et al., 1991*) plug-in in FIJI. For each channel of interest, a threshold was determined based on a negative control image and the puncta information was extracted using 'Analyze particle' function in FIJI within the region of the dendrite (outlined based on GFP signals). For the analysis of granule distribution along dendrites, the number of granules within each bin (5 µm) was determined for every dendrite, and the number in the first bin was normalized as 1.

## Immunohistochemistry, image acquisition, and quantitative analysis

For the analysis of dendritic spines in hippocampus in vivo, mouse brains were fixed at P44 and coronally sectioned at 50 µm on a vibratome (Leica). Confocal images of secondary dendrites from apical branches of CA1 hippocampal neurons and prefrontal cortex neurons were captured as described above. 3D reconstruction of individual dendrites was performed. The dendritic spine number was analyzed by Neuron Studio software.

For neuronal nuclei (NeuN), neurogranin (NRGN) and KIF5B staining, mouse brains were sacrificed at P44 and post-fixed with 4% paraformaldehyde. The samples were then sectioned to 50 µm per slice using vibratome. Brain sections were blocked with 1.5% normal goat serum (NGS) in PBST (0.3% Triton X-100) and incubated with a 1:1000 diluted primary antibody against KIF5B at 4°C overnight. Alexa 488-conjugated goat anti-rabbit IgG secondary antibody was used to probe the anti-KIF5B signals. Since both anti-KIF5B and anti-NRGN were from rabbit host, the sections were blocked again with 5% normal rabbit serum (NRS) in PBST (*Wessel and McClay, 1986*). Next, sections were incubated with 1:1000 anti-NRGN primary antibody overnight at 4°C. Another secondary antibody, goat anti-rabbit Alexa 546, was used to probe anti-NRGN signals. For NeuN staining, brain sections were incubated with anti-NeuN antibody (1:1000) after blocking. Goat anti-mouse IgG Alexa 546 conjugate was used to probe the anti-NeuN signals. Imaging was carried out under LSM700 confocal microscope. Quantification of fluorescence images was performed using ImageJ software.

## Behavioural tests

All behavioral tests were performed in the chronological order of open field test (OFT), elevated plus maze (EPM), marble burying test (MBT), 3-chamber social interaction (SI) and fear conditioning (FC). Barnes maze (BM), novel object recognition (NOR) and rotarod training were done in separate sets of animals.

Open field test. Mice were placed in the center of a square open field chamber (40 × 40 × 40 cm) surrounded by walls. Tracing was performed using ANY-maze software. The time of the mouse spent in the center area was measured over the course of 15 min (*Shin Yim et al., 2017*).

Elevated plus maze. Mice were placed in the center of a plus-shaped chamber that stands 38 cm above ground. Mice were then allowed to explore freely for 5 min. The duration of the mouse spent in either arm was recorded and tracked using ANY-maze software (*Walf and Frye, 2007*).

Marble burying test. Mice were placed into testing arenas (arena size: 42.5 cm ×27.6 cm × 15.3 cm, bedding depth: 5 cm) each containing 20 glass marbles (laid out in four rows of five marbles equidistant from one another). At the end of the 30 min exploration period, mice were carefully removed from the testing cages and the number of marbles buried was recorded. The marble burying score was arbitrarily defined as the following: four for completely buried marbles, three for marbles covered >50% with bedding, two for marbles covered 50% with bedding, one for marbles covered <50% with bedding, or 0 for anything less. The final marble burying score for each mouse was the sum of the scores of the 20 marbles (*Shin Yim et al., 2017*).

Novel object recognition. Mice were placed into a training chamber (25 cm x 25 cm x 40 cm) containing two identical objects. Mice were allowed to freely explore in the chamber for 10 min. In the recall session, mice were put back to the same chamber while one of the two identical objects were replaced with a novel object with different color and slightly different shape 14–16 hr after the training session. The movement of the mice was tracked with ANY-maze software for its interaction with both the familiar and novel objects. Discrimination index = interaction time with novel object/total interaction time with both objects (*Leger et al., 2013*).

Three-chamber social interaction. Two empty object-containment cages (shape of a cup with evenly spaced metallic bars) were each placed into the left and right chamber of a 3-chamber arena (20 cm ×42 cm × 26 cm). In the adaptation period, a mouse was shut within the center chamber for 5 min. In stage 1, a stranger mouse of same sex, similar age and size as the test mouse was put into the left cage. The test mouse in the center was released then to freely explore all of the three chambers for a 10 min period. After stage 1, the test mouse was shut within the center again when the experimenter put another stranger mouse to the right cage. At stage 2, the test mouse was allowed to explore all the three chambers again for 10 min. Approach behaviour within 2 cm with targets was defined as interaction time. Sessions were video-recorded. Approach behaviour and total distance travelled were analyzed using ANY-maze tracking system (*Shin Yim et al., 2017*).

Sociability index = (percentage time of interaction with stranger) - (percentage time of interaction with empty cage)/percentage of interaction time with both objects.

Social memory index = (percentage time of interaction with novel stranger) - (percentage time of interaction with familiar)/percentage of interaction time with both strangers.

Auditory-cued fear conditioning. FreezeFrame system (Coulbourn Instruments) was used to train and test mice. For training, the chamber was equipped with stainless-steel shocking grids, which were connecting to a precision feedback current-regulated shocker. Each chamber was contained in a sound-attenuating enclosure. Animal behaviour was recorded using low-light video cameras. Actimetrics FreezeFrame software (version 2.2; Coulbourn Instruments) was used to control the stimulus presentation by a preset program. All equipment was thoroughly cleaned with water followed by ethanol between sessions to avoid residue of scents from mouse feces and urine. Mice were habituated for 1 min on a shocking grid (cage set-up A: shocking floor grids, ethanol scent). Fear conditioning was conducted with three pairings of a 30 s, 4000 Hz, 80 dB auditory cue (CS) co-terminating with a 2 s, 0.5-mA scrambled footshock (US). The inter-trial interval was 20 s. One minute after conditioning, mice were returned to their home cages. For the recall test, mice were placed in a different context (cage set-up B: test floor grids, 1% lemon scent detergent) for an initial 2 min (pre-tone) period and this was followed by tone presentation for 2 min (CS) (*Lai et al., 2012*).

Rotarod. An EZRod system (Omnitech Electronics, Inc) was used as a motor training model. Mice were placed on the motorized rod (30 mm in diameter) in the chamber. The rotation speed gradually

increased from 0 to 100 r.p.m. over the course of 3 min. Rotarod training was performed for 20 trials, each trial lasts until the subjects dropped and the system would automatically complete that trial (*Deacon, 2013*; *Yang et al., 2014*).

Barnes maze. Mice were placed on a white circular table (92 cm in diameter, 1 m tall), which had a total of 20 holes (5 cm in diameter) separated evenly along the edge of the table. During the test, strong light with an intensity of 1500 lux and repetitive noise from metronome of 80 dB were given to serve as aversive stimuli to induce escape behaviour. On acquisition day, mice were first guided manually to the escape hole for adaptation purpose. Then, mice received 5 trials of training, with each separated from one another by 15 min. Each trial would last for 3 min. If mice were not able to find the target escape hole by the end of the each trial, mice will be guided to the target escape hole as a part of training. A 3 min recall session was carried out 5 days after acquisition day. Mice were subjected to the same maze except the escape hole was also blocked. The number of errors and latency to reach the original escape hole were measured manually to confirm the result generated by ANY-maze. Heat maps were obtained by ANY-maze. (*Sunyer et al., 2007*).

### In vivo transcranial two-photon imaging

Spine formation and elimination were examined in longitudinal studies by imaging the mouse cortex through a thinned-skull window as described previously (*Lai et al., 2012*; *Yang et al., 2009*). Briefly, one-month-old mice expressing YFP were anesthetized with ketamine/xylazine (i.p., 20 mg/ml, 3 mg/ml respectively in saline, 6 µl/g body weight). Thinned skull windows were made with high-speed microdrills in head-fixed mice. Skull thickness was reduced to about 20 µm. A two-photon microscope tuned to 920 nm (25x water immersion lens, N.A. 1.05) was used to acquire images. For re-imaging of the same region, thinned regions were identified based on the maps of the brain vasculature. Microsurgical blades were used to re-thin the region of interest until a clear image could be obtained. The area of the imaging region is 216 µm × 216 µm. The center of imaging region is located at the frontal association cortex (+2.8 mm bregma, +1.0 mm midline). All data analysis was performed blind to treatment conditions. For imaging of dendritic spines, dendritic branches were randomly sampled within a 216 µm × 216 µm area imaged at 0–100 µm distance below the pia surface. The same dendritic segments were identified from three-dimensional image stacks taken at different time points with high image quality (ratio of signal to background noise >4:1). The number and location of dendritic protrusions (protrusion lengths were more than one-third of the dendritic shaft diameter) were identified. Filopodia were identified as long, thin structures (generally larger than twice the average spine length, ratio of head diameter to neck diameter <1.2:1 and ratio of length to neck diameter >3:1). The remaining protrusions were classified as spines (*Ng et al., 2018*; *Lai et al., 2012*). The percentage of spine formation and elimination represented the number of spines formed or eliminated between the first and second view divided by the total number of spines counted at the first view in each individual mouse. For dendrite image display, fluorescent structures near and out of the focal plane of the dendrites of interest were removed manually from image stacks using Adobe Photoshop. The modified image stacks were then projected to generate two-dimensional images and adjusted for contrast and brightness.

### Statistical analysis

Data are represented as mean + SEM/SD in quantitative analysis. Statistical analysis was performed with Student's *t* test or One-way ANOVA followed by Tukey post-hoc test. If comparison was made across grouped data, Two-way ANOVA with Tukey post-hoc test was used. If dataset did not follow a normal distribution as detected by Shapiro-Wilk normality test, Mann-Whitney test or Kruskal-Wallis test with post-hoc Dunnett's multiple comparison test was used. Statistical significances were defined as $p < 0.05$.

## Acknowledgements

We are grateful to Quan Hao (University of Hong Kong) for the KIF5A and KIF5C expression constructs. We also thank Ka Ki Tam for the quantification of FMRP granule movement, the assistance of Kam Wing Kenny Ho (Chinese University of Hong Kong) on LTP recording of hippocampal slices, and the Imaging Core Facility from the Faculty of Medicine at the University of Hong Kong on image acquisition and analysis. This study was supported in part by the Research Grant Council of Hong

Kong [General Research Fund (GRF) 16100814, 17135816, 17106018 and Early Career Scheme (ECS) 27119715] awarded to KOL; the Area of Excellence Scheme (Grant AoE/M-604/16) and Theme-based Research Scheme (Grant T13-605/18 W) of the University Grants Committee of Hong Kong awarded to KOL and WHY; RGC/ECS 27103715 and RGC/GRF 17128816, National Natural Science Foundation of China (NSFC/General Program 31571031), and the Health and Medical Research Fund (HMRF) 03143096 awarded to CSWL; the Shenzhen Peacock Team Project (KQTD2015033117210153) and Shenzhen Science Technology Innovation Committee Basic Science Research Grant (JCYJ20170413154523577) awarded to JH.

## Additional information

### Funding

| Funder | Grant reference number | Author |
|---|---|---|
| Research Grants Council, University Grants Committee | GRF 16100814 | Kwok-On Lai |
| Research Grants Council, University Grants Committee | GRF 17135816 | Kwok-On Lai |
| Research Grants Council, University Grants Committee | GRF 17106018 | Kwok-On Lai |
| Research Grants Council, University Grants Committee | ECS 27119715 | Kwok-On Lai |
| University Grants Committee | AoE/M-604/16 | Wing-Ho Yung Kwok-On Lai |
| Research Grants Council, University Grants Committee | ECS 27103715 | Cora Sau Wan Lai |
| Research Grants Council, University Grants Committee | GRF 17128816 | Cora Sau Wan Lai |
| National Natural Science Foundation of China | NSFC/General Program 31571031 | Cora Sau Wan Lai |
| Health and Medical Research Fund | 03143096 | Cora Sau Wan Lai |
| Shenzhen Science and Technology Innovation Commission | Shenzhen Peacock Team Project KQTD2015033117210153 | Jiandong Huang |
| Shenzhen Science and Technology Innovation Commission | Basic Science Research Grant JCYJ20170413154523577 | Jiandong Huang |
| University Grants Committee | T13-605/18-W | Kwok-On Lai |

The funders had no role in study design, data collection and interpretation, or the decision to submit the work for publication.

### Author contributions

Junjun Zhao, Albert Hiu Ka Fok, Ruolin Fan, Pui-Yi Kwan, Hei-Lok Chan, Data curation, Formal analysis, Investigation, Visualization, Methodology; Louisa Hoi-Ying Lo, Investigation, Visualization, Methodology; Ying-Shing Chan, Methodology; Wing-Ho Yung, Formal analysis, Investigation, Methodology; Jiandong Huang, Resources, Methodology; Cora Sau Wan Lai, Kwok-On Lai, Conceptualization, Resources, Data curation, Supervision, Funding acquisition, Validation, Investigation, Visualization, Methodology, Project administration

### Author ORCIDs

Junjun Zhao (iD) https://orcid.org/0000-0001-8855-0020
Albert Hiu Ka Fok (iD) https://orcid.org/0000-0003-3553-5403
Ruolin Fan (iD) https://orcid.org/0000-0002-6405-5196

Pui-Yi Kwan (iD) http://orcid.org/0000-0001-5402-9122
Wing-Ho Yung (iD) https://orcid.org/0000-0002-5542-8173
Jiandong Huang (iD) https://orcid.org/0000-0001-7531-2816
Cora Sau Wan Lai (iD) https://orcid.org/0000-0002-5721-4259
Kwok-On Lai (iD) https://orcid.org/0000-0002-4069-054X

## Ethics

Animal experimentation: All experiments were approved and performed in accordance with University of Hong Kong Committee on the Use of Live Animals and in Teaching and Research guidelines (CULATR 3935-16 and CULATR 4056-16).

## Decision letter and Author response

Decision letter https://doi.org/10.7554/eLife.53456.sa1
Author response https://doi.org/10.7554/eLife.53456.sa2

## Additional files

### Supplementary files

- Supplementary file 1. Table for statistical tests.

- Transparent reporting form

### Data availability

All data generated or analysed during this study are included in the manuscript and supporting files.

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
