## [Decision Letter]

**Acceptance summary:**

The kinesin I family of motor proteins are known to mediate axonal transports, but whether and how they regulate dendritic transport and postsynaptic functions remains unclear. The authors report novel and differential roles of KIF5A and KIF5B, belonging to the kinesin I family, in the regulation of excitatory synapses and dendritic transports. These differential functions are mediated by the distinct C-terminal regions of KIF5A and KIF5B and mono-methylation of an arginine residue in KIF5B. Conditional deletion of KIF5B in excitatory neurons in mice leads to impairments in synaptic plasticity and learning and memory. These results suggest that members of the kinesin I family have non-redundant functions, and similar diversification and specification of functions within the motor proteins of the same family could also be observed in other motor protein families.

**Decision letter after peer review:**

[Editors’ note: the authors submitted for reconsideration following the decision after peer review. What follows is the decision letter after the first round of review.]

Thank you for submitting your work entitled "Specific depletion of the motor protein KIF5B leads to deficits in dendritic transport, synaptic plasticity and memory" for consideration by *eLife*. Your article has been reviewed by three peer reviewers, one of whom is a member of our Board of Reviewing Editors, and the evaluation has been overseen by a Senior Editor. The reviewers have opted to remain anonymous.

Our decision has been reached after consultation between the reviewers. Based on these discussions and the individual reviews below, we regret to inform you that your work will not be considered further for publication in *eLife*.

Although the reviewers find that the manuscript has many novel findings and conclusions, they raised a number of substantial concerns that need to be addressed, which would take much longer than the duration of two months that *eLife* usually allows.

Reviewer #1:

This study suggests that KIF5B is important for dendritic transport of RNA-binding proteins and the maintenance of dendritic spine density and dynamics. The manuscript has an impressive amount of data, many of the results are well controlled, and some of the data provide mechanistic details such arginine methylation. The in vivo data further support that KIF5B is important for the maintenance of spine density and dynamicity, and social and cognitive behaviors. However, it is unclear how KIF5B or KIF5B-associated FMRP contributes to the maintenance of normal spine density/dynamics. In addition, although the authors suggest that arginine methylation critically regulates KIF5B binding to FMRP, whether the KIF5B-FMRP interaction is indeed a key pathway that regulates spine maintenance/dynamics is unclear. The in vivo behavior and LTP data are quite novel, but how LTP is suppressed by KIF5B deletion is unclear.

1) Many of the data for dendritic spine density are not supported. At least, some major data such as the arginine methylation data should be supported by the electrophysiological experiments.

2) The reduced density of dendritic spines in the hippocampus in the KIF1B KO mice should be supported by mEPSC and mIPSC measurements.

3) Figure 8 tries to associate learning-related behavioral tests with the reduced hippocampal LTP. However, there is no data on basal excitatory synaptic transmissions such as input-output relationship and paired-pulse facilitation.

Reviewer #2:

The manuscript entitled "specific deletion of the motor protein KIF5B leads to dendritic transport, synaptic plasticity and memory" authored by Zhao et al. describes about the role of KIF5B in dendritic spine morphogenesis, synaptic transmission, and memory formation. The authors also showed that methylation of KIF5B mediate dendritic transport of FMRP granules and dendritic spine formation. In general, the claims made by the authors are not fully supported by the data, which is lacking in the validated experiments, data analysis and clear statistical summary. This manuscript requires more elaboration of data analysis, content, experiments and discussion.

Major points:

1) The authors need clearly describe the differential effects of KIF5A, KIF5B and KIF5C on dendritic spine morphogenesis. According to the author's results, knockdown of KIF5C also significantly affected the mushroom and thin spines (Figure 1D), indicating a potential role in synaptic transmission, similar to the KIF5B's function. It is difficult to get a clear impression on the importance of KIF5B in dendritic morphogenesis and synaptic transmission. The authors need to reorganize the manuscript to significantly strength the findings.

2) Many dendritic specific mRNAs such as CamKIIɑ have been suggested as the cargo associated with FMRP granule. Analysis of the dendritic mRNA transport and localization in KIF5B knockdown or knockout neurons would provide more evidences for supporting the role of KIF5B in this process and therefore significantly strength the author's claims.

3) FLAG-KIF5 pull-down assay (Figures 4 and 5) are absolutely suspected. Two-step FLAG beads immunoprecipitation could not provide the evidence for supporting the association of FMRP or G3BP1 with different segment of KIF5A, KIF5B and KIF5C. It would be more appropriate to use GST-fused KIF5 purification and GST pull-down assay to support the author's points.

4) It has been well established that KIF17 is essential for dendritic localization of NR2B during learning and memory. What are the effects of KIF5B knockdown or knockout on the expression of KIF17 and its cargos in dendrites? Experiments and discussion should be included to address this issue.

Reviewer #3:

The authors have made an elegant conditional mouse model revealing the importance of KIF5B for memory formation. They also show some interesting protein-protein interaction studies detailing differences between KIF5A and KIF5B tails. I agree with the authors that the isoform diversity of microtubule motors in neurons is understudied and important for the understanding of neuronal function, and this work is potentially highly significant in demonstrating some aspects of this and I support its publication. However, there is a tendency to mis-cite or exclude previous work in other model systems, which I don't believe is necessary and in fact weakens the main claims of their paper by doing so.

My main concerns are outlined below – I note that I can't offer much interpretation on whether the behavioral experiments reach standards set in the field.

Major Points:

Though there are discrepancies in the literature about whether the KLC binding region of the KHC is labelled as 'stalk' or 'tail', there is a general consensus that everything predicted to be coiled coil is termed stalk and this includes residues up to residue 758 at the least and often incorporates up to residue 810; the C-terminus of KIF5 is not typically termed "cargo binding" until after the region that binds to KLC. The second stalk coil of kinesin has been termed 'stalk' since at least Friedman and Vale, 1999, if not before. It is imperative that the authors adjust Figure 3A, Figure 4A and Figure 5A to reflect this and adjust their nomenclature throughout the manuscript to reflect this (For example the Abstract: "conserved cargo-binding domains" directly contradicts: "by their diverse carboxyl-terminal tails").

Introduction: "However, the three KIF5s have long been considered to be functionally redundant" – the cargo binding domains are not homologous, and this statement is a misrepresentation of other work in the field. Although there are studies that show some functional redundancy in some contexts (including this paper), there are others that show specificity e.g. Campbell et al., 2014. Please adjust this statement.

Discussion paragraph two: An assertion that KIF5s carboxy tails do not bind directly to cargo. This is exactly the opposite of what the papers cited show (direct binding to GRIP1 and HAP1 respectively). I also point the authors towards several other papers that show direct binding to the crago binding tail of kinesin: Xu et al., 2010; Chen and Sheng, 2013 (JCB); Randall et al., 2013 (FEBS letters); Cho et al., 2009 (EMBO reports); there are yet others. Therefore, it is imperative that the authors alter their assertion that this is a new function of the KIF5 tail.

It is unclear how reproducible the shRNA knock-down in Figure 1 really is. There are no errors bars in the supplement and no clear indication of how many times the quantification of knockdown was performed. I understand that transfection efficiency can be very low in these experiments however if this is highly variable then it has large implications for interpreting the rest of the paper. Neurons expressing GFP should be expressing the shRNA, and as the authors have isoform specific endogenous antibodies, I request quantification of knock down by quantitative immunofluorescence microscopy.

Through the cellular work in the paper, there seems to be a confusion around what should constitute a biological replicate and a technical replicate. This is most obvious in Figures 1D, 2A, 2B, 3C, 4B – measurements from multiple dendrites of one neuron are each an 'n' in the statistics reported. My criteria is typically 3 independent experiments (rather than 2 as indicated here), with the same trend obvious in all three experiments. It is context dependent on whether individual neurons in this style of experiment are considered as biological 'n' or technical replicate, and I am happy to agree that individual cells in in vitro single cell experiments (as these are) are biological replicates. However, I don't believe that different dendrites from the same neuron can be considered as additional 'n' in these experiments. The 'n' should be the number of neurons assessed, not the number of dendrites. I would appreciate the statistics to be recalculated to reflect this.

Figure 3B is very unconvincing. Were biological replicates carried out? In the blot that is shown the FMRP bands are clearly different weights. Figure 5E on the other hand is much more convincing so I strongly suggest repeating this experiment and replacing the blot. Also, the images have been aggressively cropped – I would appreciate larger view of the blot, even if just in the supplement.

Again, Figure 3C is very unconvincing. The biggest change is a tendency towards more movement with motor knock down. However, most problematic is that the kymographs don't display any of the movement highlighted underneath, which gives me major questions about what was being quantified in the first place. It appears that far more than 30% of the signal is stationary and it is difficult to make out any movement at all. The ratio numbers that appear in the text (subsection “Differential functions of KIF5A and KIF5B in dendritic transport of FMRP”) don't match the data at all and I don't understand how they were arrived at. At the very least there needs to be some work on how the kymographs are displayed. The authors’ conclusions are the opposite to mine: the changes are broadly similar, particularly looking at the lower panel of 3C.

---

## [Author Response]

[Editors’ note: the authors resubmitted a revised version of the paper for consideration. What follows is the authors’ response to the first round of review.]

Reviewer #1:

*This study suggests that KIF5B is important for dendritic transport of RNA-binding proteins and the maintenance of dendritic spine density and dynamics. The manuscript has an impressive amount of data, many of the results are well controlled, and some of the data provide mechanistic details such arginine methylation. The* in vivo *data further support that KIF5B is important for the maintenance of spine density and dynamicity, and social and cognitive behaviors. However, it is unclear how KIF5B or KIF5B-associated FMRP contributes to the maintenance of normal spine density/dynamics. In addition, although the authors suggest that arginine methylation critically regulates KIF5B binding to FMRP, whether the KIF5B-FMRP interaction is indeed a key pathway that regulates spine maintenance/dynamics is unclear. The* in vivo *behavior and LTP data are quite novel, but how LTP is suppressed by KIF5B deletion is unclear.*

We thank for the reviewer’s comments that the manuscript has an impressive amount of data and many of the results are well controlled. As pointed out by the reviewer, our present study has not elucidated how KIF5B precisely affects dendritic spine density and spine dynamics. This is because a single kinesin motor can transport diverse cargoes, including RNA-binding proteins, ion channels and scaffolding proteins. In this study we use FMRP as a dendritic cargo to demonstrate the differential binding to KIF5A and KIF5B and the consequences of their knockdown on its dendritic transport. Despite previous studies showing an important function of FMRP on dendritic spine morphogenesis, we do not think the effect of KIF5B deficiency is attributed solely to the defective transport of FMRP. Likewise, the impaired synaptic plasticity and memory of the KIF5B conditional knockout mice is likely a summative consequence of transport deficits of multiple cargoes. This notion is consistent with other studies that investigate learning and synaptic plasticity of mice deficient in a specific kinesin, in which there is no demonstration that manipulating a specific cargo of the particular kinesin can reverse the synaptic or behavioural deficits (for example, Yin et al., 2011 Neuron; Kondo et al., 2012 Neuron; Muhia et al., 2016 Cell Rep.; Gromova et al., 2018 Cell Rep.). However, without delineating the precise molecular underpinnings of how KIF5B deficiency leads to synaptic and cognitive impairment, we believe the current study is still important as it has employed multiple approaches to comprehensively demonstrate a previously unidentified specific role of KIF5B in dendritic spine morphogenesis, synaptic plasticity and cognitive function. Furthermore, we have strengthened our manuscript by performing multiple new electrophysiological experiments to characterize the altered synaptic functions of KIF5B deficient neurons according to the reviewer’s suggestions (please see below).

1) Many of the data for dendritic spine density are not supported. At least, some major data such as the arginine methylation data should be supported by the electrophysiological experiments.

We thank the reviewer for the suggestion and have performed new experiments by whole-cell patch recording on dissociated hippocampal neurons. We demonstrate that only the wild-type but not the methylation-deficient KIF5B can rescue the reduction of mEPSC frequency induced by the KIF5B-shRNA (new Figure 5G of the revised manuscript).

2) The reduced density of dendritic spines in the hippocampus in the KIF1B KO mice should be supported by mEPSC and mIPSC measurements.

We thank the reviewer for the suggestion. Accordingly, we have performed whole-cell patch recording on the CA1 neurons of acute hippocampal slices of wild-type and KIF5B conditional knockout mice. In addition to the reduction of dendritic spines, we found that the KIF5B conditional knockout neurons exhibit decrease in mEPSC frequency and amplitude as compared to the wild-type control (new Figure 7D of the revised manuscript).

3) Figure 8 tries to associate learning-related behavioral tests with the reduced hippocampal LTP. However, there is no data on basal excitatory synaptic transmissions such as input-output relationship and paired-pulse facilitation.

We have performed new experiments to compare the input-output relationship and the paired-pulse facilitation between wild-type and KIF5B conditional knockout hippocampal slices. Both properties are not significantly different across the two genotypes, indicating similar presynaptic and basal synaptic responses between the wild-type and KIF5B conditional knockout neurons. The new data has been added in the new Figure 8F-G of the revised manuscript.

Reviewer #2:[…]Major points:1) The authors need clearly describe the differential effects of KIF5A, KIF5B and KIF5C on dendritic spine morphogenesis. According to the author's results, knockdown of KIF5C also significantly affected the mushroom and thin spines (Figure 1D), indicating a potential role in synaptic transmission, similar to the KIF5B's function. It is difficult to get a clear impression on the importance of KIF5B in dendritic morphogenesis and synaptic transmission. The authors need to reorganize the manuscript to significantly strength the findings.

We thank for the reviewer’s suggestion. Although KIF5C also has a potential role in spine morphogenesis as indicated by the significant reduction in density of mushroom and thin spines after shRNA-mediated knockdown, only KIF5B-shRNA changes the percentage of different spine types by increasing the density of filopodia, thin and stubby spines. Moreover, knockdown of KIF5B resulted in significant reduction in the frequency of miniature excitatory synaptic current (mEPSC), while knockdown of KIF5C caused modest reduction in mEPSC frequency and the difference was not statistically significant from control shRNA. Therefore, knockdown of KIF5B results in more profound changes in spine morphogenesis and synaptic transmission than KIF5C knockdown. The rescue experiments show that expression of KIF5C can reverse the loss of mushroom spines induced by KIF5B-shRNA, indicating that KIF5C functions like KIF5B in promoting dendritic spine development. However, in the conditional knockout mice in which KIF5B expression is reduced but KIF5C expression remains unaffected, reduction in both spine density and mEPSC is observed in the hippocampal neurons. Therefore, the deficiency of KIF5B on synaptic function cannot be compensated by KIF5C in the brain. One possible explanation is that KIF5B is the more prominently expressed kinesin compared to KIF5C in the adult hippocampus as shown by quantitative immunoblot (Kanai et al., 2000). The presence of KIF5C may not be sufficient to compensate for the shortage of motor proteins after about 50% reduction of KIF5B expression in the conditional knockout mice. We have extended the description of the differential effects of KIF5A, KIF5B and KIF5C depletion on spine morphogenesis and mEPSC in the Results section and added the discussion on the roles of KIF5C in the revised manuscript.

2) Many dendritic specific mRNAs such as CamKIIɑ have been suggested as the cargo associated with FMRP granule. Analysis of the dendritic mRNA transport and localization in KIF5B knockdown or knockout neurons would provide more evidences for supporting the role of KIF5B in this process and therefore significantly strength the author's claims.

We thank for the reviewer’s suggestion. Accordingly we have performed fluorescence in situhybridization to examine the expression of two FMRP targets, namely CaMKIIα and Grin2b mRNAs, in dissociated hippocampal neurons after introducing shRNA targeting either KIF5A or KIF5B. Consistent with the findings of FMRP live-imaging, knockdown of KIF5B but not KIF5A significantly reduced the number of CaMKIIα and Grin2b mRNA puncta on the dendrites (new Figure 3E of the revised manuscript), thereby supporting the role of KIF5B in the dendritic targeting of FMRP and its mRNA cargoes.

3) FLAG-KIF5 pull-down assay (Figures 4 and 5) are absolutely suspected. Two-step FLAG beads immunoprecipitation could not provide the evidence for supporting the association of FMRP or G3BP1 with different segment of KIF5A, KIF5B and KIF5C. It would be more appropriate to use GST-fused KIF5 purification and GST pull-down assay to support the author's points.

We agree with the reviewer and have replaced with FLAG pull-down data by that of new GST pull-down experiments to show the differential binding of different KIF5s to various cargoes (new Figure 3B of the revised manuscript). We have kept the data on FLAG pull-down experiments in Figure 5E, because in this experiment we aim to compare the binding between wild-type and methylation-deficient mutant of KIF5B to FMRP and G3BP1. Since bacteria lack arginine methylation (Gary and Clarke, 1998 Prog. Nucleic Acid Res. Mol. Biol.), we use HEK-293T cells to express the KIF5 constructs in order to examine the effect of arginine methylation.

4) It has been well established that KIF17 is essential for dendritic localization of NR2B during learning and memory. What are the effects of KIF5B knockdown or knockout on the expression of KIF17 and its cargos in dendrites? Experiments and discussion should be included to address this issue.

We thank the reviewer for the suggestion. Accordingly, we have performed Western blot to compare the expression of KIF17 in the brains of wild-type and KIF5B conditional knockout mice. As shown in the new Figure 6—figure supplement 2, the expression of KIF17 in KIF5B conditional knockout brain is not significantly different from wild-type, indicating that the KIF5B conditional knockout does not affect KIF17 expression.

Reviewer #3:The authors have made an elegant conditional mouse model revealing the importance of KIF5B for memory formation. They also show some interesting protein-protein interaction studies detailing differences between KIF5A and KIF5B tails. I agree with the authors that the isoform diversity of microtubule motors in neurons is understudied and important for the understanding of neuronal function, and this work is potentially highly significant in demonstrating some aspects of this and I support its publication. However, there is a tendency to mis-cite or exclude previous work in other model systems, which I don't believe is necessary and in fact weakens the main claims of their paper by doing so.My main concerns are outlined below – I note that I can't offer much interpretation on whether the behavioral experiments reach standards set in the field.

We thank for the reviewer’s comments that our work is potentially highly significant. We sincerely apologize for the mis-citation of previous work, which is not intentional but mainly because we have not defined the different domains and the “carboxyl-terminus” clearly. We have taken the reviewer’s suggestion to extensively revise the text in order to clarify the definition of different domains and cited the relevant references where appropriate (please see below).

Major Points:Though there are discrepancies in the literature about whether the KLC binding region of the KHC is labelled as 'stalk' or 'tail', there is a general consensus that everything predicted to be coiled coil is termed stalk and this includes residues up to residue 758 at the least and often incorporates up to residue 810; the C-terminus of KIF5 is not typically termed "cargo binding" until after the region that binds to KLC. The second stalk coil of kinesin has been termed 'stalk' since at least Friedman and Vale, 1999, if not before. It is imperative that the authors adjust Figure 3A, Figure 4A and Figure 5A to reflect this and adjust their nomenclature throughout the manuscript to reflect this (For example the Abstract: "conserved cargo-binding domains" directly contradicts: "by their diverse carboxyl-terminal tails").

We are grateful to the reviewer’s suggestions. In Figure 3A of the original manuscript, the schematic diagram denoting various KIF5 domains was made according to the study by Setou et al., 2002. In that study, the “cargo-binding domain” of KIF5A is defined as residue 807-934 based on the binding to GRIP1, and this region is very conserved among KIF5A, 5B and 5C (more than 80% amino acid identity). In contrast, the “diverse carboxyl-terminal tail” in our manuscript specifically refers to the region beyond residue 934 until the last amino acid. This “tail” region, which determines the functional specificity revealed by our study, is highly diverse among the three KIF5s. Therefore in our original manuscript the “conserved cargo-binding domain” and “diverse carboxyl-tail” refer to different and non-overlapping regions, and they together constitute the “tail domain” defined by Friedman and Vale, 1999. We apologize for the poor definition of different regions in the original manuscript that caused the confusion. In line with the reviewer’s recommendation, we now adopt the definition of the different domains based on Friedman and Vale, 1999, and revise the schematic diagram (new Figure 3A) to correct the positions of the stalk domain (until residue 801 of KIF5A, which includes the second coiled-coil) and the tail domain, as well as adding the KLC-binding domain (a.a. 771-801 for KIF5A). To avoid confusion, we have now used the term “carboxyl-terminus” throughout the revised manuscript to define the diverse regions of the three KIF5s (a.a. 934-1027 for KIF5A; a.a. 936-963 for KIF5B; and a.a. 937-956 for KIF5C; new Figure 3A, 4A, 5A) and to distinguish them from the “tail domain” defined by Friedman and Vale.

Introduction: "However, the three KIF5s have long been considered to be functionally redundant" – the cargo binding domains are not homologous, and this statement is a misrepresentation of other work in the field. Although there are studies that show some functional redundancy in some contexts (including this paper), there are others that show specificity e.g. Campbell et al., 2014. Please adjust this statement.

We have now modified the statement to “Functional redundancy has been demonstrated among the three KIF5s, as exogenous expression of KIF5A or KIF5C can rescue the impaired mitochondrial transport in cells lacking KIF5B (Kanai et al., 2000). […] Furthermore, only KIF5A dysfunction leads to seizure and the neuromuscular disorder Hereditary Spastic Paraplegia (Fink, 2013; Nakajima et al., 2012).”.

Discussion paragraph two: An assertion that KIF5s carboxy tails do not bind directly to cargo. This is exactly the opposite of what the papers cited show (direct binding to GRIP1 and HAP1 respectively). I also point the authors towards several other papers that show direct binding to the crago binding tail of kinesin: Xu et al., 2010; Chen and Sheng, 2013 (JCB); Randall et al., 2013 (FEBS letters); Cho et al., 2009 (EMBO reports); there are yet others. Therefore, it is imperative that the authors alter their assertion that this is a new function of the KIF5 tail.

We apologize again for the confusion here. As pointed out in the earlier response, the “carboxyl tail” in our original manuscript specifically refers to residues 936 to 963 of KIF5B, which represents the carboxyl-terminal portion of a larger “tail” defined by Friedman and Vale, 1999. We now re-name it as “carboxyl-terminus” in the revised manuscript, and we think that this region (KIF5B936-963) does not bind directly to cargo because GST-KIF5B constructs lacking this region still pull down multiple cargoes (e.g. Setou et al., 2002 Nature; Cho et al., 2007 Traffic; Xu et al., 2010 J. Neurosci.; Barry et al., 2014 Dev. Cell; Lin et al., 2019 iScience). Our present study also suggests that this carboxyl-terminus is not for cargo binding because removing it (a.a. 940-1027) from KIF5A increases, rather than decreases, the pull-down of FMRP (Figure 4A). However, substituting this carboxyl terminus of KIF5A by that of KIF5B is sufficient to transform KIF5A into a kinesin motor that enhances mushroom spine morphogenesis (Figure 4B). We therefore define the carboxyl-terminus (KIF5B936-963) as a crucial determinant for functional specificity, which to our knowledge has not been reported and represents a new function for this region. We have included this discussion on the functional significance of the carboxyl-terminus in the revised manuscript.

It is unclear how reproducible the shRNA knock-down in Figure 1 really is. There are no errors bars in the supplement and no clear indication of how many times the quantification of knockdown was performed. I understand that transfection efficiency can be very low in these experiments however if this is highly variable then it has large implications for interpreting the rest of the paper. Neurons expressing GFP should be expressing the shRNA, and as the authors have isoform specific endogenous antibodies, I request quantification of knock down by quantitative immunofluorescence microscopy.

We thank for the reviewer’s suggestion. We have now performed three independent experiments to knockdown down individual KIF5 in cortical neurons by nucleofection, and the quantification was statistically analyzed to show the knockdown efficiency and specificity (new Figure 1—figure supplement 1A). In addition, we have confirmed the knockdown efficiency of KIF5B-shRNA by immunofluorescence staining after introduction into hippocampal neurons by immunofluorescence microscopy (new Figure 1—figure supplement 1B).

*Through the cellular work in the paper, there seems to be a confusion around what should constitute a biological replicate and a technical replicate. This is most obvious in Figures 1D, 2A, 2B, 3C, 4B – measurements from multiple dendrites of one neuron are each an 'n' in the statistics reported. My criteria is typically 3 independent experiments (rather than 2 as indicated here), with the same trend obvious in all three experiments. It is context dependent on whether individual neurons in this style of experiment are considered as biological 'n' or technical replicate, and I am happy to agree that individual cells in* in vitro *single cell experiments (as these are) are biological replicates. However, I don't believe that different dendrites from the same neuron can be considered as additional 'n' in these experiments. The 'n' should be the number of neurons assessed, not the number of dendrites. I would appreciate the statistics to be recalculated to reflect this.*

We thank and agree with the reviewer’s suggestion that biological replicates should be defined as individual neurons. Accordingly, we have performed additional experiments such that all the relevant analyses of dendritic spines and FMRP or RNA granules in primary neurons are quantification from three independent experiments, and the “n” is the number of neurons instead of number of dendrites. The quantification method is now clarified in the Materials and methods section of the revised manuscript. The recalculated sample sizes and statistical analyses are also included in the figure legend of the new Figure 1D, 2A, 2B, 3D, 3E and 4B.

Figure 3B is very unconvincing. Were biological replicates carried out? In the blot that is shown the FMRP bands are clearly different weights. Figure 5E on the other hand is much more convincing so I strongly suggest repeating this experiment and replacing the blot. Also the images have been aggressively cropped – I would appreciate larger view of the blot, even if just in the supplement.

As described in the response to reviewer #2 (major point 3), we have repeated the pull-down experiments by performing new GST-pull down experiments to show the differential binding of different KIF5s to the cargoes, and the size of the FMRP bands pulled down by different KIF5s are consistent. As a result, the original Figure 3B has now been replaced (new Figure 3B in the revised manuscript).

Again, Figure 3C is very unconvincing. The biggest change is a tendency towards more movement with motor knock down. However, most problematic is that the kymographs don't display any of the movement highlighted underneath, which gives me major questions about what was being quantified in the first place. It appears that far more than 30% of the signal is stationary and it is difficult to make out any movement at all. The ratio numbers that appear in the text (subsection “Differential functions of KIF5A and KIF5B in dendritic transport of FMRP”) don't match the data at all and I don't understand how they were arrived at. At the very least there needs to be some work on how the kymographs are displayed. The authors’ conclusions are the opposite to mine: the changes are broadly similar, particularly looking at the lower panel of 3C.

We thank for the reviewer’s comments. Our observation that most of the FMRP granules are stationary or oscillatory with little movement is consistent with previous studies on the movement of RNA granules by Kosik’s laboratory (Knowles et al., 1996 J. Neurosci.; Rook et al., 2000 J. Neurosci.) and RNA-binding proteins (Mitsumori et al., 2017). Those relatively immotile FMRP granules mask the minority of granules that undergo greater movement. We have now replaced the representative images with those that show clearer movement of granules and indicate the different types of granules with lines of different colours. We have explicitly defined the movement of the five different types of granules and added the detailed definition in the Materials and methods section of the revised manuscript. We have also performed additional experiments and re-analyzed the data. The new live-imaging data (new Figure 3C-D) indicates that the number of FMRP granules in the dendrites is significantly reduced after knockdown of KIF5B but not KIF5A. The new data also shows that KIF5B knockdown specifically reduces the number of stationary granules. Since other kinesins besides KIF5 can also bind to FMRP (Charalambous et al., 2013; Davidovic et al., 2007), we speculate that the different pools of FMRP granules are carried by different KIFs, with KIF5B mainly responsible for the less motile granules and while other KIFs transport the more motile pools of FMRP. The immobility of RNA granules on dendrites might be important for their local translation (Doyle and Kiebler, 2011 EMBO J), and it is possible that besides a conventional transport function, KIF5B may help to anchor the dendritically localized FMRP and mRNAs near synapses for local translation in response to extracellular stimulus. The preferential effect on stationary granules after KIF5B knockdown has been discussed in the revised manuscript. To further support the role of KIF5B in dendritic transport of FMRP, we have performed new in situ hybridization experiments to demonstrate the reduced FMRP mRNA cargoes (CaMKIIα and Grin2b) in dendrites after knockdown of KIF5B but not KIF5A (new Figure 3E).